# Pre- and postsynaptic upregulation of FasII synergistically underlies neuropathological and behavioral phenotypes in a *Drosophila* model of myotonic dystrophy

Alex Chun Koon [1,13], Ka Yee Winnie Yeung[1,13], Yitao Wu[1], Lok I Leong[1], John Tsun Po Cheung[1], Zhefan Stephen Chen [1,2], Shaohong Isaac Peng[1], Noah S. Armstrong [3], C. Andrew Frank [3], Paul Magneron [4], Mário Gomes-Pereira [4], Joyce Man See Fung[1], Ariadna Bargiela[5,6], Nerea Moreno[5,6,7], Javier Poyatos-Garcia[5,6,7,8], Juan Vilchez [5,7,8], Aline Huguet-Lachon[4], Cassandra Kussius Brewer[9], Max Zinter[9], Erin S. Beck [10], Rubén Artero [5,6,7], Genevieve Gourdon [4], Vivian Budnik [9], Travis Thomson [9], Brian D. McCabe [11] & Ho Yin Edwin Chan [1,2,12] ✉

Myotonic dystrophy type 1 is a multisystemic disorder that has been extensively studied for decades, yet our understanding of its neuropathological aspect remains rudimentary. Building on an established *Drosophila* model, we study the neuropathological features of the disease by expressing untranslated expanded *CUG* repeats at the *Drosophila* larval neuromuscular junction. In this model, we show that both pre- and postsynaptic expressions of *CUG* repeats participate in inducing phenotypes in synaptic boutons, arbors, transmission and larval locomotor activity. Furthermore, expression of *CUG* repeats in either motorneurons or body wall muscles induces upregulation of the cell adhesion molecule FasII (NCAM1 in mammals), and the knockdown of *fasII* is sufficient to rescue the phenotypes. Overexpression of FasII-C, a FasII isoform with no cytoplasmic domain, mimics the phenotypes of expanded *CUG* expression at the neuromuscular junction. In contrary, overexpression of FasII-A-PEST+ rescues the synaptic and behavioral defects. Our study provides insights into the fundamental mechanisms underlying synapse dysregulation in myotonic dystrophy type 1.

Myotonic dystrophy type 1 (DM1), also known as Steinert disease, is the most common form of adult-onset muscular dystrophy, affecting as many as one in every 2100 individuals[1]. This multisystemic disorder is characterized not only by progressive myotonia and muscle degeneration but also by cataracts, heart dysfunction, and neuropathology[2]. Congenital DM1 can also occur in infants and children, who present with severe muscle weakness and hypotonia rather than myotonia, as well as cognitive impairment[3]. DM1 is caused by a *CTG* trinucleotide repeat expansion in the 3′ UTR of the *dystrophia myotonica protein kinase* (*DMPK*) gene[4–6]. Normal individuals may have fewer than 37 *CTG* repeats, whereas DM1 patients may have hundreds or even thousands of repeats[7]. The gain-of-function from mRNA transcripts harboring these expansions of untranslated *CUG* repeats was found to contribute to some of the major pathological features of DM1 independently of

the *DMPK* locus[8]. In fact, the severity of DM1 was found to correlate with the number of repeats and age of onset[9]. *CUG* repeat-containing transcripts were shown to be retained in the nucleus and recruited into ribonuclear foci[10], leading to the sequestration of RNA-binding proteins such as muscleblind-like 1 (MBNL1) and subsequently compromising the RNA-splicing machinery[11,12].

RNA toxicity is well-known to be associated with the neurodegenerative features of numerous repeat expansion diseases, including polyglutamine diseases, spinocerebellar ataxias, and C9ORF72-associated amyotrophic lateral sclerosis/frontotemporal dementia[13]. DM1 is no exception. Despite its original designation as a muscular dystrophy, DM1 is known to manifest many neuropathological features[14–16]. DM1 patients commonly exhibit degeneration of the pigmentary retina and loss of photoreceptor neurons[2]. They also may exhibit cognitive impairment, speech and language difficulties, attention deficit, autism spectrum disorder, autistic features, sleep disorder, social anxiety, and peripheral neuropathy[17–23]. Nevertheless, the neuropathological aspect of DM1 is much less explored than the muscle pathology. In neurons, mutant mRNAs containing *CUG* repeats were found to accumulate in ribonuclear foci within the nuclei and to co-localize with MBNL1, similar to the process observed in muscles[12]. However, in a transgenic mouse model of DM1 (DM300), an expansion of 300 *CUG* repeats did not induce detectable neuropathology[24]. In contrast, an expansion of 1300 repeats in DMSXL mice was found to result in motor neuropathy, suggesting that a large *CUG* expansion can indeed result in neuropathology[25]. The end-plates at the neuromuscular junctions (NMJs) of DMSXL mice exhibit decreased size and complexity, and 23% are disconnected from the axonal branches; furthermore, the motorneurons exhibit defects in conducting action potentials[25]. Synaptic proteins such as RAB3A and synapsin I are respectively upregulated and hyperphosphorylated in DMSXL mice, in association with synaptic transmission and behavioral deficits[15]. Neuronal progeny cells derived from human embryonic stem cells carrying the DM1 mutation exhibit defects in neurite outgrowth and synapse formation[16]. Despite these findings, the mechanism underlying the influence of expanded *CUG* repeats on neurons and synapses remains unclear.

*Drosophila melanogaster* has emerged as a valuable model for studying the molecular pathology of DM1 in vivo. Transgenic flies expressing expanded *CUG* repeats in muscles recapitulate hallmark features of both the mouse DM1 model and human disease, including the formation of RNA foci, misregulation of alternative splicing, hypercontraction of muscles, splitting of fibers, progressive degeneration of muscle, and genome-wide changes in expression that are both splice-dependent and -independent[26–28]. These models provide a genetically tractable platform for dissecting pathogenic mechanisms and identifying potential therapeutic targets.

Here, we used the highly stereotypical and accessible *Drosophila* larval NMJ system to study the effects of untranslated *CUG* repeats on synaptic functions and behavior. We found that the simultaneous expression of expanded *CUG* in the presynaptic motorneurons (MNs) and postsynaptic body wall muscles (BWMs) resulted in synergistic functional and structural impairment of the NMJ. We characterized this DM1 model of neuropathology and identified the upregulation of Fasciclin II (FasII, a.k.a. Fas2), the *Drosophila* orthologue of mammalian neural cell adhesion molecule 1 (NCAM1), as a major contributor to the observed NMJ phenotypes. We observed dysregulation of NCAM1 in both the DMSXL mouse model and patients with DM1, supporting the relevance of our findings in *Drosophila*. Overexpression of FasII-C, an isoform of FasII with no cytoplasmic domain, mimicked the phenotypes of expanded *CUG* expression at the NMJ. We further rescued our DM1 model by either knocking down the upregulated *fasII* or overexpressing the major neuronal isoform of FasII at the NMJ, suggesting that synaptic loss might be alleviated by restoring the proper ratios of cell adhesion molecule isoforms. These findings provide important

insights into the mechanisms underlying motor defects associated with DM1 neuropathology.

## Results

### Simultaneous pre- and postsynaptic expression of expanded *CUG* repeats causes neuromuscular phenotypes

We used the *Drosophila* larval NMJ system to examine how *CUG* repeat-mediated toxicity affected neuronal growth and development. The larval NMJ is characterized by a high level of plasticity during development and large, easily visualized synapses that produce easily recordable data[29]. Importantly, this system also allows easy manipulation of the presynaptic MNs and postsynaptic BWMs separately. *C380-Gal4 (C380)* and *C57-Gal4 (C57)* are, respectively presynaptic MN and postsynaptic BWM driver lines. Using these Gal4 drivers, we overexpressed either a control construct (*UAS-CTG_{60}*) containing fewer untranslated *CUG* repeats than the disease threshold or an untranslated expanded *CUG* repeat construct (*UAS-CTG_{480}*), which was previously characterized in an adult *Drosophila* DM1 model[27,30]. Fluorescence in situ hybridization showed that $CUG_{480}$ expression in MNs and BWMs of 3rd instar larvae resulted in RNA foci containing *CUG* repeats in the nuclei of these cells (Supplementary Fig. 1). Compared with the $CUG_{60}$ control, the expression of $CUG_{480}$ in either MNs or BWMs alone did not result in any change in synaptic bouton numbers in late 3rd instar larvae. However, a robust bouton reduction phenotype was observed when both the *C57* and *C380* drivers were used to drive $CUG_{480}$ expression in the pre- and postsynaptic compartments simultaneously (Fig. 1a, b). This phenotype was not observed when we overexpressed an untranslated *CAG* repeat expansion construct (*UAS-CAG_{250}*)[31] using both drivers, suggesting that the bouton phenotype is specific to expanded *CUG* repeats (Fig. 1b).

This phenotypic reduction in boutons could have been caused by reduced synaptic growth during development, the disassembly/retraction of mature boutons after they were formed or a combination of both. If the phenotype is present in early larval stages, then it is likely to be due to reduced synaptic growth during development. Thus, we dissected 1st and 2nd instar larvae expressing $CUG_{480}$ via C380 and C57 drivers to investigate whether they had reduced bouton numbers. Interestingly, we found that both the 1st and 2nd instar larvae had no detectable reductions in bouton numbers, whereas 3rd instar larvae exhibited a robust decrease (Fig. 1c). These results suggest that the NMJs of $CUG_{480}$-expressing animals were morphologically normal during the early larval stages, with the bouton reduction phenotype appearing only around the 3rd instar stage.

Checking for the presence of mature postsynaptic markers after synaptic disassembly/retraction would be a more direct way to determine whether this process is present in $CUG_{480}$-expressing larvae. Discs large (DLG) is the *Drosophila* counterpart of the mammalian postsynaptic MAGUK scaffolding proteins, which include SAP-97, SAP-70, and PSD-95[32,33], and it is only present in the postsynaptic density of a mature bouton. Through a detailed analysis of the NMJs, we identified some disconnected, disassembling boutons from seemingly degenerating arbors in larvae expressing $CUG_{480}$ using both the *C380* and *C57* drivers (Fig. 1d, e). We observed faint expression of DLG on the BWMs near these disassembling boutons, indicating that these were once mature boutons with postsynaptic densities (Fig. 1d). Thus, these disassembling boutons differ from ghost boutons (i.e., premature boutons that lack DLG on postsynaptic BWMs[34]). Although we cannot rule out the possibility that the bouton reduction phenotype in these animals is due to reduced synaptic growth during development, we are certain that the disassembly of mature boutons and arbors contributed to this phenotype, at least in part. Interestingly, although the expression of $CUG_{480}$ using the presynaptic MN driver (C380) alone did not induce a detectable bouton reduction phenotype (Fig. 1b), it resulted in a small but significant increase in arbor disassembly (Fig. 1e). However, the expression of $CUG_{480}$ using both the

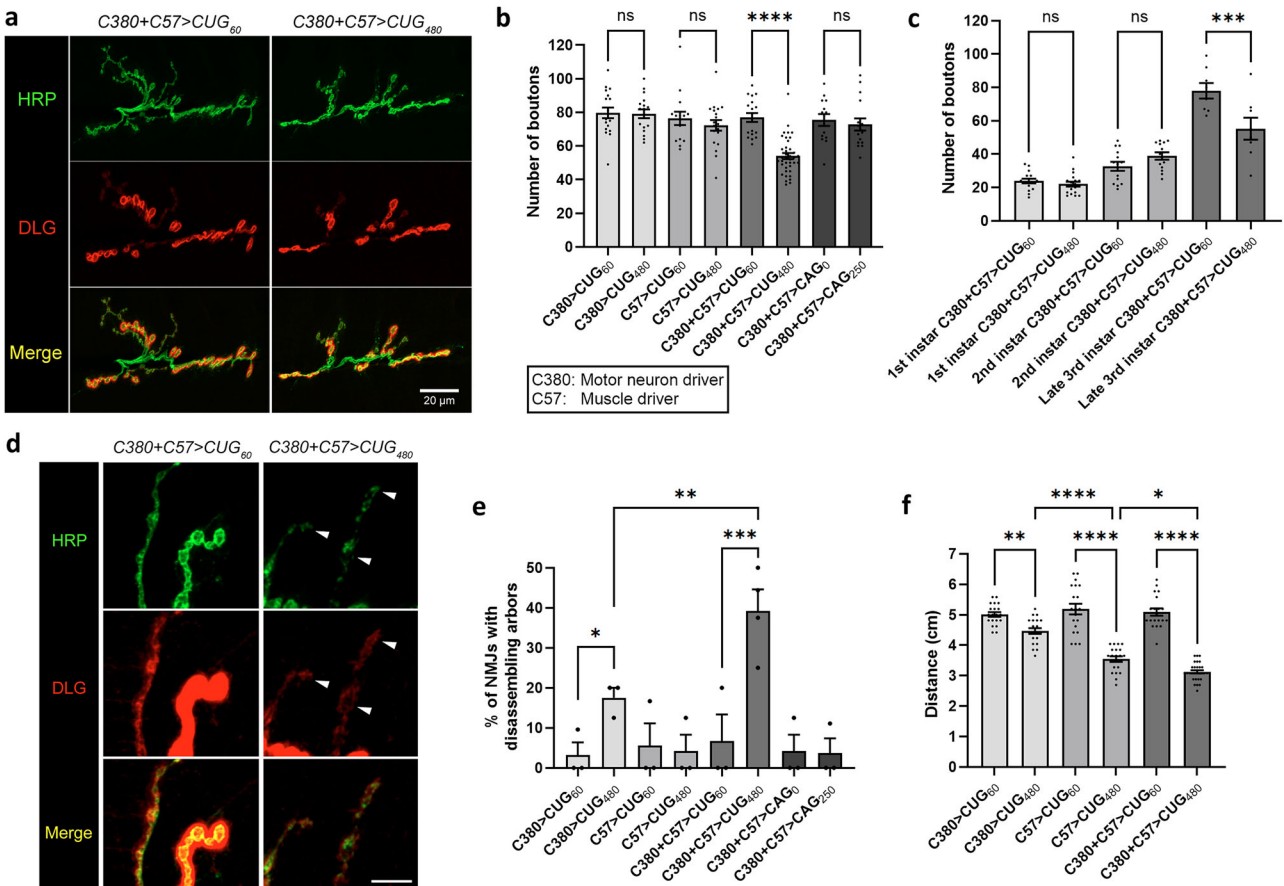

**Fig. 1 | Simultaneous pre- and postsynaptic overexpression of $CUG_{480}$ causes NMJ and locomotor defects. a** Confocal micrographs of *Drosophila* NMJs of late 3rd instar larvae at muscles 6 and 7 of segment A3. Anti-HRP (in green) marks the presynaptic boutons. Anti-Discs large (DLG) (in red) marks the postsynaptic density. Scale bar is 20 μm. **b** Quantification of bouton numbers in (**a**) and in other related genotypes. Untranslated $CAG_0$ and $CAG_{250}$ were used as additional controls to show sequence specificity of the phenotype observed. $n = 18, 17, 15, 19, 18, 39, 14, 16$, where $n$ is the number of analyzed NMJs. **c** Quantification of bouton numbers in 1st, 2nd and late 3rd instar larvae overexpressing either $CUG_{60}$ or $CUG_{480}$ using *C380* and *C57*. $n = 16, 18, 14, 14, 10, 10$, where $n$ is the number of analyzed NMJs.

**d** Confocal micrographs at high-magnification showing arbors of boutons of NMJs at muscles 6 and 7 of segment A3. White arrowheads denote signs of disassembling boutons and arbors. Scale bar is 5 μm. **e** Quantification of disassembling arbors in (**d**) and in other related genotypes. $N = 3$, with each $N$ is a set of NMJs analyzed from at least 4 larvae, and no more than two NMJs were analyzed per larva. **f** Quantification of larval locomotor activity. $n = 20, 20, 20, 20, 20, 30$, where $n$ indicates the number of analyzed larvae. Each larva is defined as a biological replicate, and no more than two NMJs were analyzed per larva. Analysis of variance (one-way ANOVA) with Tukey post-hoc test was performed. Histograms depict mean ± SEM. *$p < 0.05$, **$p < 0.01$, ***$p < 0.001$, ****$p < 0.0001$.

presynaptic (*C380*) and postsynaptic (*C57*) drivers resulted in a significantly higher amount of arbor disassembly than using the *C380* driver alone (Fig. 1e). These results suggest that although the presynaptic expression of $CUG_{480}$ alone does not yield a bouton number phenotype, it does cause morphological defects. Our results also suggest that the presence of *CUG* RNA toxicity in both the presynaptic MNs and postsynaptic BWMs has a synergistic effect on arbor disassembly.

Structural changes at the NMJ are usually accompanied by functional changes. Thus, after analyzing the morphology of the NMJs, we explored whether the expression of $CUG_{480}$ would also affect larval crawling behavior. A previous study demonstrated that the expression of expanded *CUG* repeats in postsynaptic BWMs alone was sufficient to induce locomotor defects in *Drosophila* larvae[28]. As the *C380*-driven expression of $CUG_{480}$ in MNs alone was sufficient to cause arbor disassembly (Fig. 1e), we hypothesized that it may also be sufficient to cause locomotor defects in larvae. Indeed, we found that *C380*-driven expression of $CUG_{480}$ alone induced a small but significant locomotor defect in 3rd instar larvae, while expression of $CUG_{480}$ using the postsynaptic driver (*C57*) alone was sufficient to produce a stronger locomotor defect; the expression of $CUG_{480}$ using both the *C380* and

*C57* drivers produced the strongest phenotype (Fig. 1f). These results suggest that *CUG* RNA toxicity in both presynaptic MNs and postsynaptic BWMs contributes to a robust behavioral defect in larva.

To better understand how expanded *CUG* RNA affects NMJ functions, we examined synaptic transmission in *CUG*-expressing animals through electrophysiological analyses. The control $CUG_{60}$ or expanded $CUG_{480}$ was expressed presynaptically (*C380*), postsynaptically (*C57*), or pre+postsynaptically (*C380* + *C57*), and the miniature excitatory postsynaptic potential (mEPSP) and excitatory postsynaptic potential (EPSP) were measured. Expression of $CUG_{480}$ in presynaptic MNs led to increases in mEPSP and EPSP amplitudes (Fig. 2a, b, and e) but no detectable changes in quantal content (QC) or mEPSP frequency (Fig. 2c–e). Expression of $CUG_{480}$ in postsynaptic BWMs led to decreases in mEPSP and EPSP amplitudes (Fig. 2f, g, and j) but no detectable changes in QC or mEPSP frequency (Fig. 2h–j). Expression of $CUG_{480}$ pre+postsynaptically resulted in no detectable changes in mEPSP amplitude (Fig. 2k, o), possibly because the increasing effect of presynaptic expression (Fig. 2a) and decreasing effect of postsynaptic expression (Fig. 2f) cancelled each other. However, pre+postsynaptic expression of $CUG_{480}$ actually caused a significant increase in EPSP amplitude (Fig. 2l, o), despite the slight decreasing effect due to

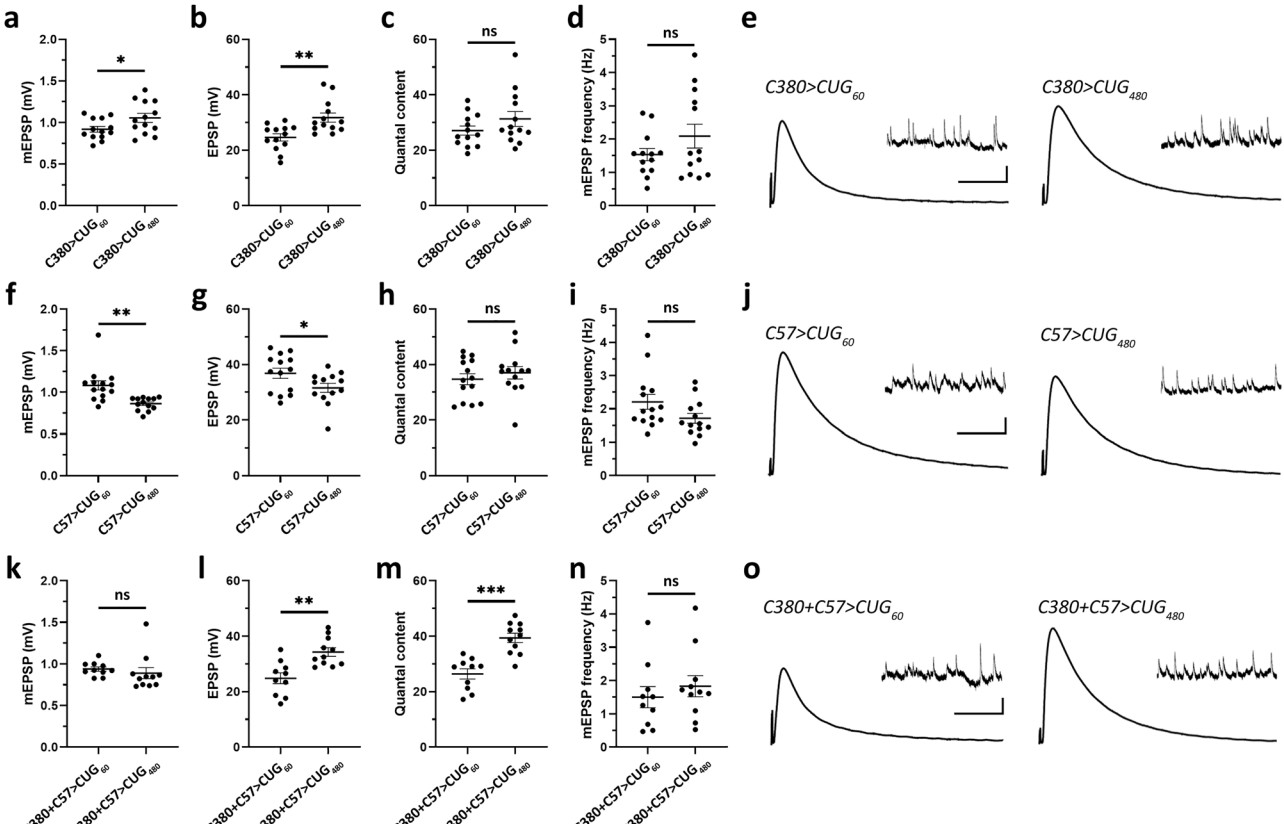

**Fig. 2 | Pre- and postsynaptic expression of $CUG_{480}$ alters synaptic functions at the *Drosophila* larval NMJ. a–e** Electrophysiology data collected from *Drosophila* larval NMJs expressing $CUG_{60}$ or $CUG_{480}$ presynaptically using *C380-Gal4*. $n = 13,13$, where n is the number of analyzed NMJs. **f–j** Electrophysiology data collected from *Drosophila* larval NMJs expressing $CUG_{60}$ or $CUG_{480}$ postsynaptically using *C57-Gal4*. $n = 14,13$, where n is the number of analyzed NMJs. **k–o** Electrophysiology data collected from *Drosophila* larval NMJs expressing $CUG_{60}$ or $CUG_{480}$ pre +postsynaptically using *C380 + C57*. $n = 10,11$, where $n$ is the number of analyzed NMJs. **a, f, k** mEPSP amplitude. **b, g, l** EPSP amplitude. **c, h, m** Quantal content. **d, i, n** mEPSP frequency. **e, j, o** Representative electrophysiological traces. Scale bar for EPSP (mEPSP): $y = 5$ mV (1 mV), $x = 50$ ms (2 s). Student's (two-tailed) *t*-test was used. Histograms depict mean ± SEM. *$p < 0.05$, **$p < 0.01$, ***$p < 0.001$. Each larva is defined as a biological replicate, and no more than two NMJs were analyzed per larva.

postsynaptic expression (Fig. 2g). Interestingly, although no change in QC was detected when $CUG_{480}$ was expressed either pre- or post-synaptically (Fig. 2c, h), simultaneous pre+postsynaptic expression of $CUG_{480}$ resulted in a significant increase in QC, indicating a synergistic effect (Fig. 2m). This increase in QC may indicate an increased release of vesicles per evoked response. In contrast, no detectable change in mEPSP frequency was observed when $CUG_{480}$ was expressed pre + postsynaptically (Fig. 2n, o).

In summary, simultaneous pre- and postsynaptic expression of expanded *CUG* repeats synergistically induced morphological and functional phenotypes at the larval NMJ and locomotor behavioral defects. These results have helped to establish the *Drosophila* larval NMJ as a model for studying neuropathology in DM1. As expanded *CUG* repeats must be expressed in both pre- and postsynaptic components to induce certain phenotypes, the ability to flexibly manipulate pre- and postsynaptic cells in the NMJ provided us with a prime system for dissecting the neuropathology underlying DM1.

## FasII/NCAM1 is upregulated in both DM1 models and patients with DM1

We next sought to determine the cause of the phenotypes in our DM1 model. Although many molecules can alter bouton numbers at the NMJ, few are present in both presynaptic MNs and postsynaptic BWMs and regulate bouton numbers synergistically. A potential candidate is the cell adhesion molecule FasII, the *Drosophila* orthologue of mammalian NCAM1. FasII-mediated regulation of bouton numbers at the

NMJ is complex. First, the overexpression of total FasII presynaptically, postsynaptically, or both can lead to different bouton number phenotypes[35]. Second, *fasII* null mutant-induced lethality can only be rescued by expressing FasII in both the central nervous system (CNS) and BWMs[36,37], indicating that it plays both key pre- and postsynaptic roles. Most importantly, *fasII* hypomorphic mutants exhibit a synaptic retraction phenotype[36,37], which is similar to the arbor disassembly phenotype we observed in our DM1 NMJ model (Fig. 1d, e). Therefore, we hypothesized that expanded *CUG* RNA might dysregulate FasII.

To test this hypothesis, we expressed $CUG_{480}$ in the CNS and muscles and collected the respective tissues for mRNA or protein analysis. The MN driver *C380* was not suitable for this purpose, as its expression was limited to only a few neurons in the CNS. Instead, we used *elav^{GeneSwitch}-Gal4* (*elav^{GS}*), an inducible pan-neuronal Gal4 driver expressed only in the presence of RU486[38]. This GeneSwitch driver was used to avoid expressing $CUG_{480}$ throughout the CNS at a too-early stage of embryonic development, as its expression in the CNS was shown to affect viability[27]. First, we verified that the *elav^{GS} + C57* model was also sufficient to cause NMJ bouton phenotypes, while *elav^{GS}* alone was insufficient (Supplementary Fig. 2). Then, we simultaneously expressed $CUG_{480}$ in the CNS and muscles of larva using *elav^{GS}* and *C57*, reared the animals in standard *Drosophila* medium containing 50 μM RU486 since hatching, dissected the CNS and BWMs at the 3rd instar larval stage, and performed semi-quantitative RT-PCR to determine the total *fasII* levels in these tissues. Our results demonstrated the upregulation of *fasII* transcripts in both types of tissues (Fig. 3a–d).

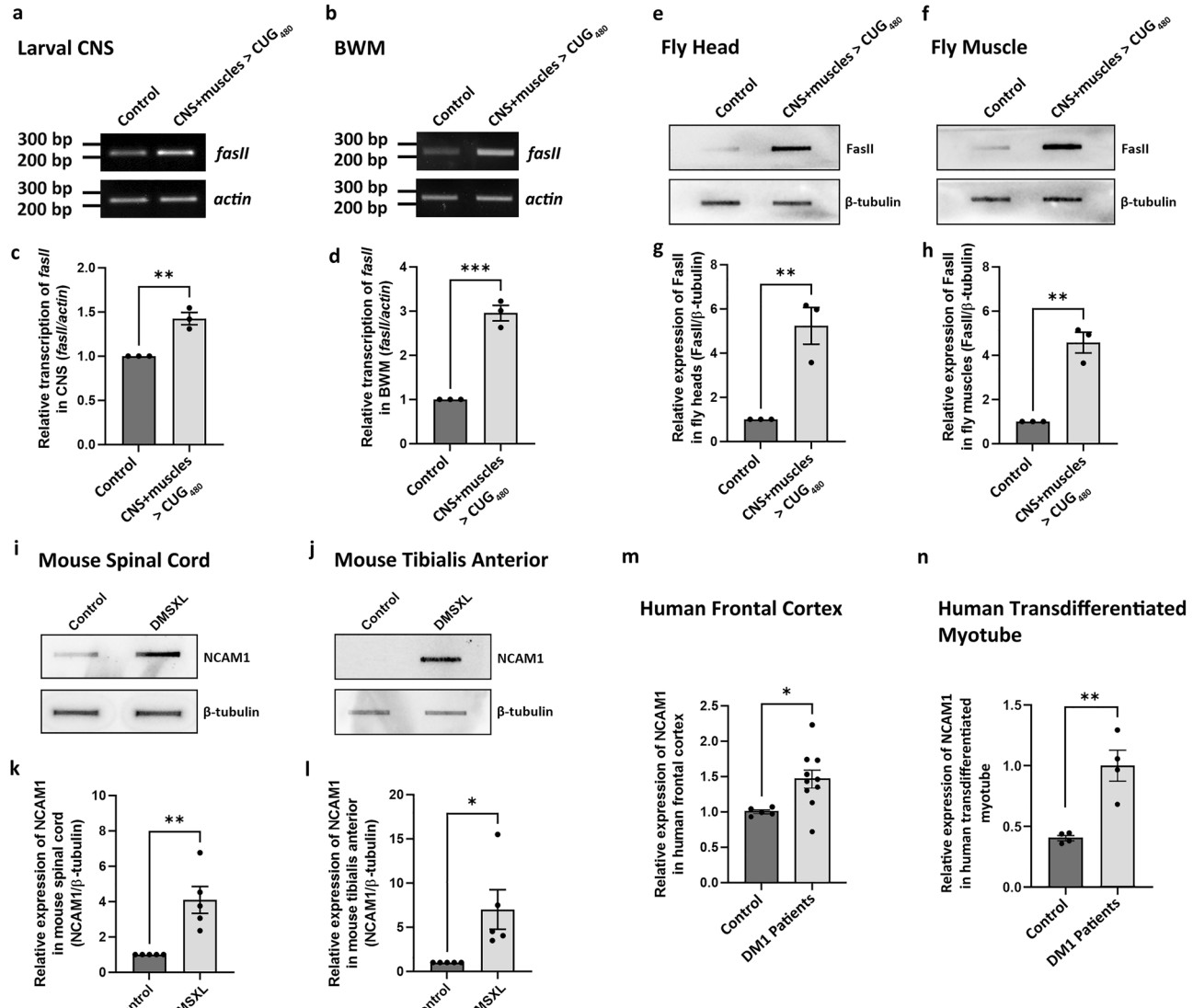

**Fig. 3 | FasII/NCAM is upregulated in DM1 models and DM1 patients.**
**a**, **b** Representative semi-quantitative RT-PCR of *fasII* in *Drosophila* larval (**a**) CNS and (**b**) BWM. **c** Quantification of (**a**). $N = 3$ for both genotypes. **d** Quantification of (**b**). $N = 3$ for both genotypes. **e**, **f** Representative slot blots of FasII in adult *Drosophila* (**e**) head and (**f**) muscle. **g** Quantification of (**e**). $N = 3$ for both genotypes. **h** Quantification of (**f**). $N = 3$ for both genotypes. **i**, **j** Representative slot blots of NCAM1 in mouse (**i**) spinal cord and (**j**) muscle. **k** Quantification of (**i**). $N = 3$ for both genotypes. **l** Quantification of (**j**). $N = 3$ for both genotypes. **m**, **n** The relative expression level of NCAM1 protein in human frontal cortex (CNS) and myotube (muscle) tissues when comparing the DM1 patients to healthy control subjects. **m** Quantified slot blot data of NCAM1 in human frontal cortex. $N = 5$ for control. $N = 10$ for DM1 patients. **n** Quantitative dot blot data of NCAM1 in human transdifferentiated myotube. $N = 4$ for both control and DM1 patients. Each $N$ is a biological replicate. Student's *t*-test (two-tailed) was used. Histograms depict mean ± SEM. *$p < 0.05$, **$p < 0.01$, ***$p < 0.001$.

To show that this upregulation was not limited to the larval stage, we allowed these animals to grow to the adult stage, collected the fly heads and muscles, and performed slot blot to detect total FasII proteins. We observed significant upregulation of FasII in both the CNS and muscles of $CUG_{480}$-expressing adult flies (Fig. 3e–h). This result suggests that both *fasII* transcript and overall FasII protein expression are upregulated by the expanded *CUG* repeats, and this upregulation persists from the larval stages to adulthood.

To investigate the presence of a similar upregulation of cell adhesion molecules in a mammalian model of DM1, we analyzed neural and skeletal muscle samples from DMSXL mice, a well-characterized model of DM1[15,39,40]. Using the slot blot technique, we found that NCAM1 protein was indeed upregulated in the spinal cord and tibialis anterior of DMSXL mice (Fig. 3i–l), confirming that our findings in Drosophila are relevant to mammals.

To further verify that the upregulation of FasII in *Drosophila* and of NCAM1 in mice is relevant to humans with DM1, we detected human NCAM1 by performing slot blots using the frontal cortex of either control individuals or patients with DM1 and quantitative dot blots using transdifferentiated myotubes derived from the myoblasts of controls or patients with DM1. We again observed the upregulation of NCAM1 protein in these human tissues (Fig. 3m, n), confirming that our findings are truly relevant to patients with DM1.

**Knockdown of *fasII* rescues NMJs in the *Drosophila* DM1 model**
As DM1 is known to be associated with splicing machinery defects[11,12], we wondered whether particular *fasII* isoforms were upregulated in our *Drosophila* model. At least four known isoforms of FasII are expressed in *Drosophila*: the two transmembrane isoforms, FasII-A-PEST+ and FasII-A-PEST−, are the major isoforms in neurons[41]. FasII-C

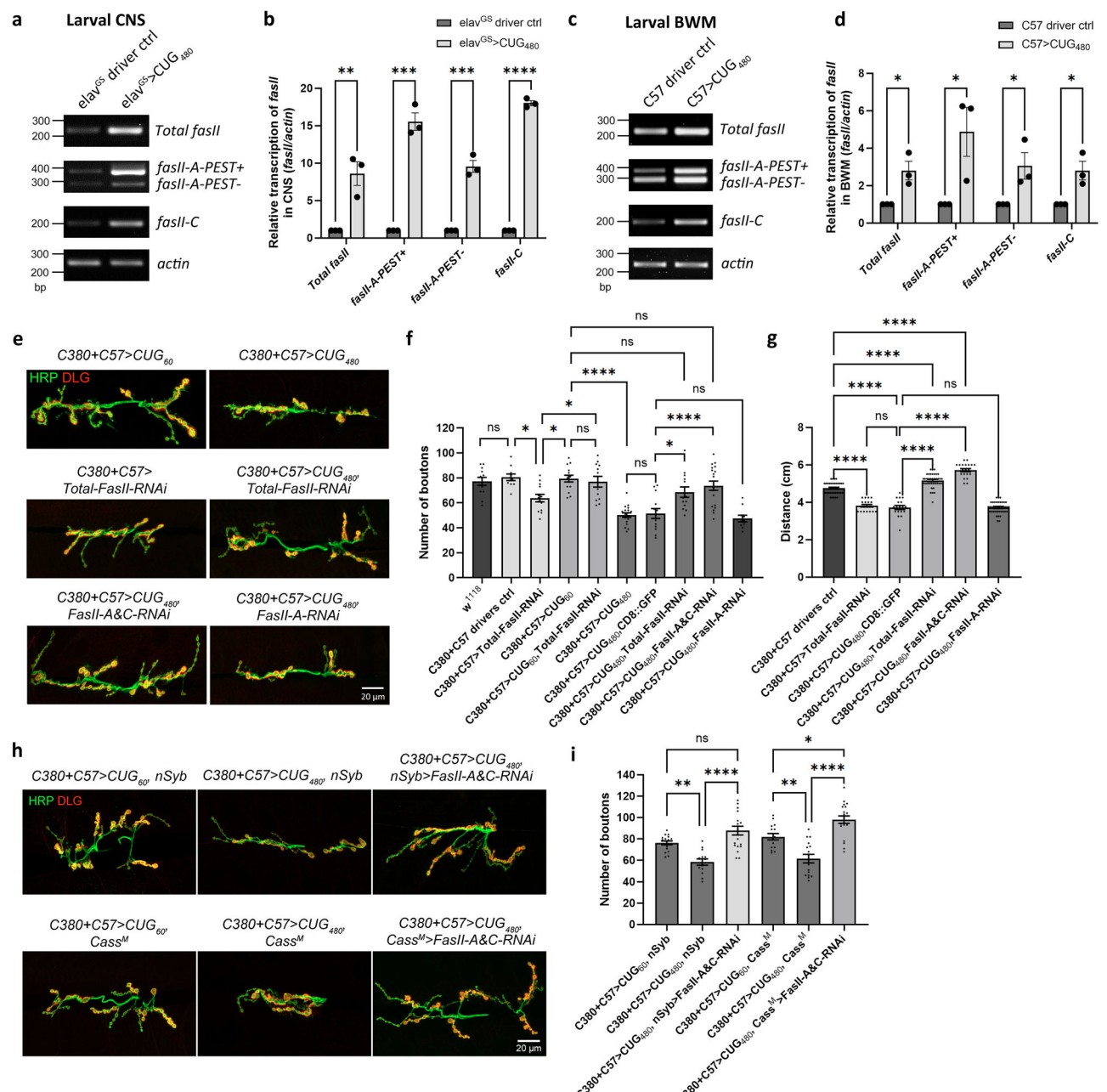

**Fig. 4 | Knockdown of *fasII* rescues NMJs in the *Drosophila* DM1 model.**
**a–d** Semi-quantitative RT-PCR of specific *fasII* isoforms in *Drosophila* larval.
**a** Representative images of RT-PCR performed using larval CNS. **b** Quantification of
(**a**), *N* = 3. Each *N* is an independent experiment and is defined as a biological
replicate. **c** Representative images of RT-PCR performed using larval BWM.
**d** Quantification of (**c**), *N* = 3. Each *N* is an independent experiment and is defined as
a biological replicate. **e** Confocal micrographs of *Drosophila* NMJs of late 3rd instar
larvae at muscles 6 and 7 of segment A3. Anti-HRP (in green) marks the presynaptic
boutons. Anti-DLG (in red) marks the postsynaptic density. Scale bar is 20 μm.
**f** Quantification of bouton numbers in (**e**) and in other related genotypes. *n* = 12, 10,

13, 14, 12, 18, 13, 13, 19, 10, where *n* is the number of analyzed NMJs. **g** Quantification
of larval locomotor activity. *n* = 40, 20, 18, 40, 20, 40, where *n* indicates the number
of analyzed larvae. **h** Confocal micrographs of *Drosophila* NMJs of late 3rd instar
larvae at muscles 6 and 7 of segment A3. Scale bar is 20 μm. **i** Quantification of
bouton numbers in (**h**). *n* = 17, 14, 20, 15, 16, 20, where *n* is the number of analyzed
NMJs. Student's *t*-test (two-tailed) was used in (**b** and **d**). Each larva is defined as a
biological replicate, and no more than two NMJs were analyzed per larva. One-way
ANOVA with Tukey post-hoc test was performed in (**f**, **g**, and **i**). Histograms depict
mean ± SEM. \**p* < 0.05, \*\**p* < 0.01, \*\*\*\**p* < 0.0001.

has no transmembrane or cytoplasmic domain and is tethered to the
cell membrane via a GPI anchor[42]. FasII-B is a less-characterized iso-
form lacking a clear transmembrane domain and a GPI anchor[43]. To
investigate the transcript expression of *fasII* isoforms, we over-
expressed $CUG_{480}$ using either *elav^GS* or *C57* and reared the animals in
either medium containing 50 μM RU486 or plain medium, then dis-
sected either the CNSs or BWMs at the late 3rd instar larval stage. Semi-
quantitative RT-PCR was performed to analyze the transcript levels of

*fasII-A-PEST+*, *fasII-A-PEST−*, and *fasII-C* isoforms. All three *fasII* iso-
forms were upregulated in both the CNS and BWMs (Fig. 4a–d).

If upregulated FasII expression in MNs and BWMs caused the NMJ
morphological defects and larval crawling behavioral defects observed
our DM1 model, then we should be able to rescue these phenotypes by
reducing FasII expression in these tissues using RNAi. We thus knocked
down *fasII* using a previously characterized *UAS-Total-FasII-RNAi*
line[44,45]. First, we tested the effect of expressing *UAS-Total-FasII-RNAi*

using the presynaptic *C380* and postsynaptic *C57* drivers on a control background and observed a small but significant decrease in synaptic bouton numbers (Fig. 4e, f). Second, we co-expressed *UAS-Total-FasII-RNAi* and $CUG_{60}$ (*UAS-CTG_{60}*) and found that, intriguingly, $CUG_{60}$ nullified the bouton reduction effect caused by *UAS-Total-FasII-RNAi* (Fig. 4f). Third, we sought to determine whether *UAS-Total-FasII-RNAi* could rescue the bouton reduction phenotype caused by $CUG_{480}$. However, one caveat of this experiment was that the extra *UAS* expressed by the RNAi construct might reduce the Gal4 available for $CUG_{480}$ expression, resulting in a false rescue. Thus, we introduced *UAS-CD8::GFP* to control for the extra *UAS* in the rescue genotype. We observed no significant difference in bouton numbers between $C380 + C57 > CUG_{480}$ and $C380 + C57 > CUG_{480}$, *CD8::GFP*, indicating that the extra *UAS* transgene expression did not significantly affect the bouton reduction phenotype caused by $CUG_{480}$ (Fig. 4f). We then co-expressed *UAS-Total-FasII-RNAi* and $CUG_{480}$ (*UAS-CTG_{480}*) and found that *fasII* knockdown rescued the bouton numbers (Fig. 4e, f).

To further examine whether the knockdown of particular *fasII* isoforms was responsible for the observed rescue, we used two fly lines designed to knock down specific *fasII* isoforms. The previously characterized *UAS-FasII-A-RNAi* can knock down both *fasII-A-PEST+* and *fasII-A-PEST−*[44]. *UAS-FasII-C-RNAi* was designed to target exon 8 of *fasII*, which encodes the GPI anchor. Thus, it was expected to knock down only *fasII-C*. However, RT-PCR analysis of *Drosophila* BWMs expressing this construct revealed that it actually knocked down both the *fasII-A-PEST+* and *fasII-A-PEST−* isoforms, in addition to *fasII-C* (Supplementary Figs. 3 and 4). Therefore, we renamed this line *UAS-FasII-A&C-RNAi*. When either *UAS-FasII-A-RNAi* or *UAS-FasII-A&C-RNAi* was expressed at the NMJ on the wild-type background, we observed reductions in bouton numbers and locomotor activity (Supplementary Fig. 5). When we knocked down *fasII-A* and *fasII-C* simultaneously using *UAS-FasII-A&C-RNAi* in $CUG_{480}$-expressing animals, we observed a rescue of bouton numbers, similar to that observed with *UAS-Total-FasII-RNAi* (Fig. 4e, f). In contrast, *fasII-A* knockdown alone could not rescue the bouton numbers in $CUG_{480}$-expressing animals (Fig. 4e, f). Our data suggest that the rescue in our DM1 model was not due to the knockdown of *fasII-A* isoforms.

As *fasII* knockdown rescued the bouton numbers at the NMJ, we further explored whether this manipulation could also rescue larval crawling behavior. Consistent with our findings regarding bouton numbers, we found that the expression of *UAS-Total-FasII-RNAi* and *UAS-FasII-A&C-RNAi* under the *C380 + C57* drivers could rescue crawling behavior in DM1 model animals, whereas *FasII-A-RNAi* was incapable of rescue (Fig. 4g). Intriguingly, the expression of *UAS-Total-FasII-RNAi* and *UAS-FasII-A&C-RNAi* on the DM1 background led to an over-rescue phenotype in locomotor activity.

In Fig. 1, we show that the expression of $CUG_{480}$ in either presynaptic MNs or postsynaptic BWMs alone was not sufficient to reduce bouton numbers. Hypothetically, rescuing either the presynaptic MNs or postsynaptic BWMs via *fasII* knockdown should be sufficient to rescue bouton numbers at the NMJ in our DM1 model. To test this hypothesis, we needed to perform tissue-specific *fasII* knockdown independent of the Gal4/UAS system, as $CUG_{480}$ was overexpressed by *C380* and *C57* simultaneously in our model. We thus used the LexA/LexAop binary expression system in conjunction with the Gal4/UAS system[46]. *nSyb-LexA (nSyb)* is a pan-neuronal driver line expressed in all neurons, while *Cass^M-LexA (Cass^M)* is a muscle driver line expressed in BWMs (Supplementary Fig. 6). We generated *LexAop-FasII-A&C-RNAi* using the same construct used in *UAS-FasII-A&C-RNAi*. Similar to our findings with *UAS-FasII-A&C-RNAi*, we found that *LexAop-FasII-A&C-RNAi* also knocked down both the *fasII-A* and *fasII-C* isoforms (Supplementary Fig. 7). We then expressed *LexAop-FasII-A&C-RNAi* to knock down *fasII* using either *nSyb* or *Cass^M* on the $C380 + C57 > CUG_{480}$ DM1 model background. As expected, *fasII-A* and *fasII-C* knockdown using either *nSyb* or *Cass^M* was sufficient to rescue the bouton numbers in

our DM1 model (Fig. 4h, i). In fact, knockdown of *fasII-A* and *fasII-C* using *Cass^M* resulted in an over-rescue phenotype. These results suggest that the upregulation of FasII by $CUG_{480}$ in this DM1 model must be present in both the presynaptic MNs and postsynaptic BWMs to cause a decrease in bouton number at the NMJ.

## Overexpression of FasII-C results in NMJ morphological defects and behavioral defects that strongly resemble the DM1 model

At this point, we had accumulated several observations. First, a previous study suggested that FasII-C is usually not the major isoform expressed in neurons[41]. Thus, its upregulation in $CUG_{480}$-expressing MNs and BWMs may negatively impact NMJ functions. Second, *UAS-Total-FasII-RNAi* and *UAS-FasII-A&C-RNAi*, but not *UAS-FasII-A-RNAi*, could rescue bouton numbers and behavior (Fig. 4e–g), suggesting that FasII-C upregulation may have caused the NMJ defects observed in our DM1 model. To test this possibility, we overexpressed *UAS-FasII-A-PEST+*, *UAS-FasII-A-PEST−*, or *UAS-FasII-C* both pre- and postsynaptically by using the *C380* and *C57* drivers simultaneously (on the $w^{1118}$ control background). We found that although overexpression of FasII-A-PEST+ or FasII-A-PEST− did not affect bouton numbers, overexpression of FasII-C reduced bouton numbers, similar to the DM1 model (Fig. 5a, b). Overexpression of FasII-C also led to an increase in arbor disassembly (Fig. 5c, d), another phenotype observed when we overexpressed $CUG_{480}$ at the NMJ (Fig. 1d, e). This phenotype was not observed with FasII-A-PEST+ or FasII-A-PEST− overexpression (Fig. 1d).

Regarding larval crawling behavior, pre- and postsynaptic overexpression of FasII-A-PEST+ resulted in a significant increase in locomotor activity, while overexpression of FasII-A-PEST− had no impact and overexpression of FasII-C resulted in a decrease in locomotor activity (Fig. 5e). Again, overexpression of FasII-C mimicked the phenotype of the DM1 model.

In Fig. 1, we show that simultaneous pre- and postsynaptic overexpression of $CUG_{480}$ was required for a reduction in bouton numbers at the NMJ (Fig. 1b). We explored whether there was a similar requirement underlying the effects of FasII-C overexpression on the NMJ. Indeed, overexpression of FasII-C using a single driver (either *C380* or *C57*) was insufficient to induce significant changes in bouton numbers at the NMJ (Fig. 5f, g), indicating a synergistic effect between pre- and postsynaptic expression of FasII-C on NMJ morphology. Overexpression of FasII-C in BWMs alone (using *C57*) was also insufficient to induce arbor disassembly, while overexpression of FasII-C in MNs alone (using *C380*) slightly but significantly increased arbor disassembly (Fig. 5h), similar to that observed with $CUG_{480}$ overexpression (Fig. 1e). Presynaptic overexpression of FasII-C resulted in a slight but significant decrease in locomotor activity, while postsynaptic overexpression of FasII-C caused a further decrease; pre + postsynaptic overexpression caused the most severe decrease in mobility (Fig. 5i). These phenotypes also strongly resemble those of $CUG_{480}$ overexpression (Fig. 1f).

## Simultaneous pre- and postsynaptic overexpression of FasII-C synergistically induces transmission phenotypes at the NMJ

To further evaluate the effects of FasII-C overexpression on NMJ functions, we performed electrophysiological analyses to examine synaptic transmission. We overexpressed *UAS-FasII-C* using either *C380*, *C57* or both drivers simultaneously and measured the mEPSPs and EPSPs. We found that presynaptic or postsynaptic overexpression of FasII-C alone had no impact on mEPSP amplitude (Fig. 6a, b, m, n). In contrast, simultaneous pre- and postsynaptic overexpression of FasII-C resulted in a significant decrease in mEPSP amplitude, indicating a decrease in either vesicle size or glutamate receptor expression at the NMJ (Fig. 6c, o). Presynaptic or postsynaptic overexpression of FasII-C resulted in a small but significant decrease in EPSPs, while simultaneous pre- and postsynaptic overexpression resulted in a further decrease (Fig. 6d–f, m–o), indicating an overall decrease in the evoked

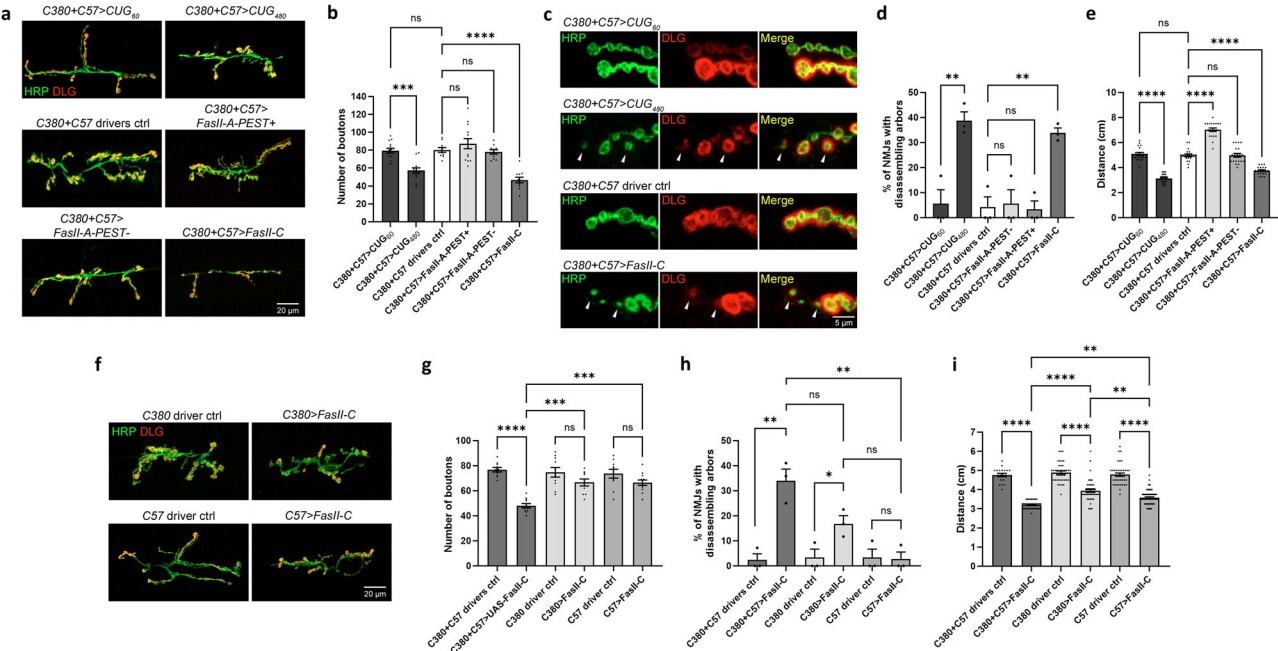

**Fig. 5 | Overexpression of FasII-C causes in NMJ morphological defects and larval locomotor impairment. a** Confocal micrographs of *Drosophila* NMJs of late 3rd instar larvae at muscles 6 and 7 of segment A3. Anti-HRP (in green) marks the presynaptic boutons. Anti-DLG (in red) marks the postsynaptic density. Scale bar is 20 μm. **b** Quantification of bouton numbers in (**a**). *n* = 14, 13, 10, 13, 10, 11, where *n* is the number of analyzed NMJs. **c** Confocal micrographs at high-magnification showing arbors of boutons of NMJs at muscles 6 and 7 of segment A3. White arrowheads denote signs of disassembling boutons and arbors. Scale bar is 5 μm. **d** Quantification of disassembling arbors in (**c**). *N* = 3, with each *N* is a set of NMJs analyzed from at least 4 larvae, and no more than two NMJs were analyzed per larva. **e** Quantification of larval locomotor activity. *n* = 20, 20, 30, 20, 20, 20, where *n*

indicates the number of analyzed larvae. **f** Confocal micrographs of *Drosophila* NMJs of late 3rd instar larvae at muscles 6 and 7 of segment A3. Scale bar is 20 μm. **g** Quantification of bouton numbers in (**f**). *n* = 10, 11, 10, 10, 10, 12 where *n* is the number of analyzed NMJs. **h** Quantification of disassembling arbors. *N* = 3, with each *N* is a set of NMJs analyzed from at least 4 larvae, and no more than two NMJs were analyzed per larva. **i** Quantification of larval locomotor activity. *n* = 80, 19, 40, 40, 40, 80, where *n* indicates the number of analyzed larvae. Each larva is defined as a biological replicate, and no more than two NMJs were analyzed per larva. One-way ANOVA with Tukey post-hoc test was performed. Histograms depict mean ± SEM. \**p* < 0.05, \*\**p* < 0.01, \*\*\**p* < 0.001, \*\*\*\**p* < 0.0001.

response. Presynaptic overexpression had no impact on QC, while postsynaptic overexpression resulted in a small but significant decrease and simultaneous pre- and postsynaptic overexpression resulted in a further decrease (Fig. 6g–i), indicating a decrease in the number of vesicles released per evoked response. Lastly, in terms of mEPSP frequency, presynaptic overexpression resulted in a small but significant decrease, postsynaptic overexpression had no effect and simultaneous pre- and postsynaptic overexpression caused a further decrease (Fig. 6j–o), indicating a decrease in the number of spontaneously released vesicles. All these data strongly suggest that simultaneous presynaptic and postsynaptic overexpression of *FasII-C* synergistically impairs NMJ functions.

### Overexpression of FasII-C exacerbates and FasII-A rescues NMJ phenotypes of the *Drosophila* DM1 model

As our data suggest that FasII-C upregulation is the major cause of the NMJ phenotypes of our *Drosophila* DM1 model, we wondered whether the overexpression of FasII-C on our disease model background would worsen the phenotypes and whether overexpression of FasII-A isoforms would ameliorate the phenotypes. To test whether FasII-C expression would exacerbate the pathological phenotypes of our disease model, we co-expressed *CUG480* and *UAS-FasII-C* using *C380* and *C57* and compared the resulting animals with those co-expressing *CUG480* and *UAS-CD8::GFP*. Animals co-expressing *CUG480* and *UAS-CD8::GFP* displayed a bouton reduction phenotype when comparing with *CUG60*-expressing control animals, while those co-expressing *CUG480* and *UAS-FasII-C* exhibited an even stronger bouton reduction phenotype (Fig. 7a, b). Similarly, animals co-expressing *CUG480* and *UAS-FasII-C* exhibited more strongly impaired larval locomotor activity

than those co-expressing *CUG480* and *UAS-CD8::GFP* (Fig. 7c). These results suggest that the upregulation of FasII-C in the DM1 model may not have reached the saturation point, such that a further increase in FasII-C expression resulted in more severe phenotypes.

To test whether overexpression of FasII-A isoforms could ameliorate the pathological phenotypes of our disease model, we overexpressed either *UAS-FasII-A-PEST+* or *UAS-FasII-A-PEST−* in conjunction with *CUG480* (*UAS-CTG480*) using *C380* and *C57*. Intriguingly, overexpression of either of the two FasII-A isoforms was sufficient to rescue the bouton numbers in our DM1 model (Fig. 7d–f). Moreover, overexpression of *UAS-FasII-A-PEST+* also rescued locomotor impairment in our DM1 model (Fig. 7g), even resulting in over-rescue. However, overexpression of *UAS-FasII-A-PEST−* did not lead to an observable improvement of locomotor activity (Fig. 7g). To determine why *UAS-FasII-A-PEST−* could not rescue locomotor activity, we analyzed the bouton morphology in *UAS-FasII-A-PEST−* animals using high-magnification confocal microscopy, and found the boutons displaying abnormalities (Fig. 7h, i). These morphological abnormalities might have contributed to the lack of behavioral rescue in the *UAS-FasII-A-PEST−* animals.

## Discussion

DM1 is a widely recognized multisystemic disorder with neurological manifestations, including both peripheral nervous system and CNS abnormalities[14]. Despite decades of studies, however, the underlying neuropathology remains one of the most poorly understood aspects of this disease. In this study, we utilized an established *Drosophila* model of DM1 and studied the neuropathological features of DM1 by expressing untranslated expanded *CUG* repeats at the *Drosophila*

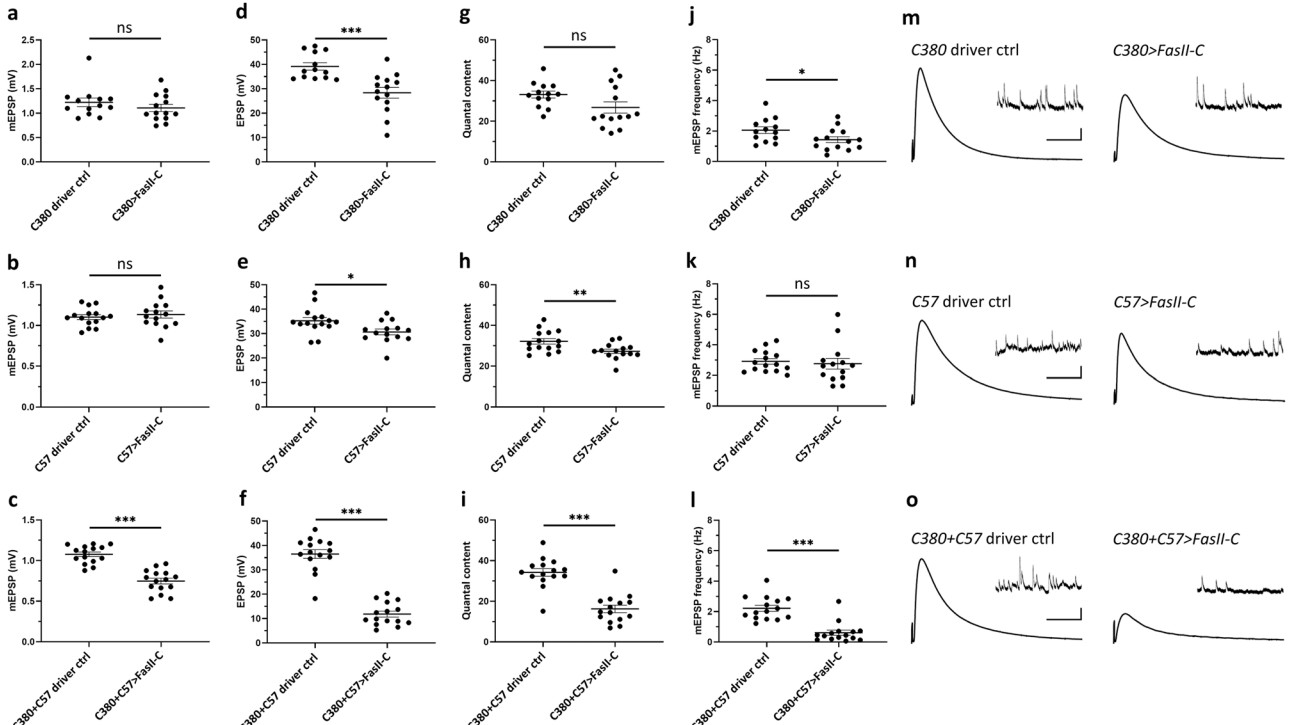

**Fig. 6 | Simultaneous pre- and postsynaptic overexpression of FasII-C synergistically impairs synaptic functions at the *Drosophila* larval NMJ. a–c** mEPSP amplitude of *Drosophila* larval NMJ overexpressing FasII-C using (**a**) *C380*, **b** *C57*, **c** *C380* + *C57*. **d–f** EPSP amplitude of *Drosophila* larval NMJ overexpressing FasII-C using (**d**) *C380*, **e** *C57*, **f** *C380* + *C57*. **g–i** Quantal content of *Drosophila* larval NMJ overexpressing FasII-C using (**g**) *C380*, **h** *C57*, **i** *C380* + *C57*. **j–l** mEPSP frequency of *Drosophila* larval NMJ overexpressing FasII-C using (**j**) *C380*, **k** *C57*, **l** *C380* + *C57*.

**m–o** Representative electrophysiological traces. Scale bar for EPSP (mEPSP): $y = 5$ mV (1 mV), $x = 50$ ms (2 s). For (**a**, **d**, **g**, **j**), $n = 13, 14$, where $n$ is the number of analyzed NMJs. For (**b**, **e**, **h**, **k**), $n = 15, 14$, where n is the number of analyzed NMJs. For (**c**, **f**, **i**, **l**), $n = 15, 15$, where $n$ is the number of analyzed NMJs. Each larva is defined as a biological replicate, and no more than two NMJs were analyzed per larva. Student's *t*-test (two-tailed) was used. Histograms depict mean ± SEM. *$p < 0.05$, **$p < 0.01$, ***$p < 0.001$.

larval NMJ. We observed a synaptic bouton reduction phenotype at the NMJ that only occurred when $CUG_{480}$ was simultaneously expressed in presynaptic MNs and postsynaptic BWMs (Fig. 1a, b). Both pre- and postsynaptic expression of $CUG_{480}$ also contributed to an arbor disassembly phenotype (Fig. 1d, e), larval locomotion impairment (Fig. 1f), and synaptic transmission phenotypes (Fig. 2). We determined that $CUG_{480}$ expression induced upregulation expression of the cell adhesion molecule FasII at the NMJ (Fig. 3a–f). Similar upregulation of the orthologous NCAM1 was observed in a mouse model of DM1 and in patients with DM1 (Fig. 3g–n). Knocking down *fasII* rescued the reduced bouton and abnormal locomotor phenotypes in our *Drosophila* DM1 model (Fig. 4f, g). We further found that pre- and postsynaptic overexpression of FasII-C mimicked the NMJ morphological and behavioral phenotypes observed in the DM1 model (Fig. 5) and synergistically induced synaptic transmission defects (Fig. 6). Finally, we demonstrated that FasII-C overexpression exacerbated the NMJ and locomotor phenotypes in the DM1 model (Fig. 7a–c), whereas overexpression of either of the two FasII-A isoforms rescued bouton numbers in the model (Fig. 7d–f). The FasII-A isoforms FasII-A-PEST+ was even capable of rescuing the locomotor phenotype in the DM1 model (Fig. 7g).

In the animal kingdom, cell adhesion molecules play a crucial role in coordinating cell–cell interactions and provide navigational cues during nervous and muscular system development. In *Drosophila*, the roles of FasII in axon guidance and neuronal development during embryogenesis have been extensively studied[36,37]. FasII directs axon fasciculation through homophilic cell–cell recognition to establish and organize a scaffolding foundation for the developing nervous system[42]. FasII-A-PEST+ (a.k.a. Fas2-RA) and FasII-A-PEST− (a.k.a. Fas2-RC) have been identified as the major isoforms at the larval NMJ, with

FasII-A-PEST+ being the predominant species expressed in neurons[47]. All FasII isoforms contain five Ig-like domains that can mediate adhesion via transhomophilic binding (Supplementary Fig. 8a). However, only FasII-A-PEST+ and FasII-A-PEST− include a transmembrane domain connected to a cytoplasmic PDZ-interacting domain that can participate in intracellular signaling (Supplementary Fig. 8a). The cytoplasmic domains of the two FasII-A isoforms were previously shown to interact with DLG at the postsynapse[48,49] and thus may facilitate retrograde signals required for synapse maintenance[50]. In contrast, FasII-C (a.k.a. Fas2-RB) lacks an intracellular cytoplasmic domain and is attached to the plasma membrane via a GPI anchor instead (Supplementary Fig. 8a). Functionally, FasII-C was proposed to act as a homotypic bridging protein within renal Malpighian tubule cells to stabilize the microvillar brush border against shear stress[51]. In *Drosophila* expressing expanded *CUG* repeats, abnormal upregulation of FasII-C may cause intracellular signals to be dampened at the NMJ, which in turn may compromise synapse integrity. Our results show that overexpression of either FasII-A isoform rescued bouton numbers in $CUG_{480}$-expressing animals (Fig. 7d–f). Possibly, the overexpressed FasII-A outcompeted the upregulated FasII-C in the disease model and restored the proper ratio of FasII isoforms at the NMJs, subsequently restoring the intracellular signals required to maintain synapse integrity.

Despite many similarities between the NMJ and behavioral phenotypes associated with $CUG_{480}$ and *UAS-FasII-C* overexpression, obvious differences in the synaptic transmission phenotypes were observed. Pre + postsynaptic expression of $CUG_{480}$ increased the EPSP amplitude and QC (Fig. 2), whereas pre + postsynaptic expression of *UAS-FasII-C* decreased the mEPSP and EPSP amplitudes, QC and mEPSP frequency (Fig. 6). These discrepancies can be explained by the fact

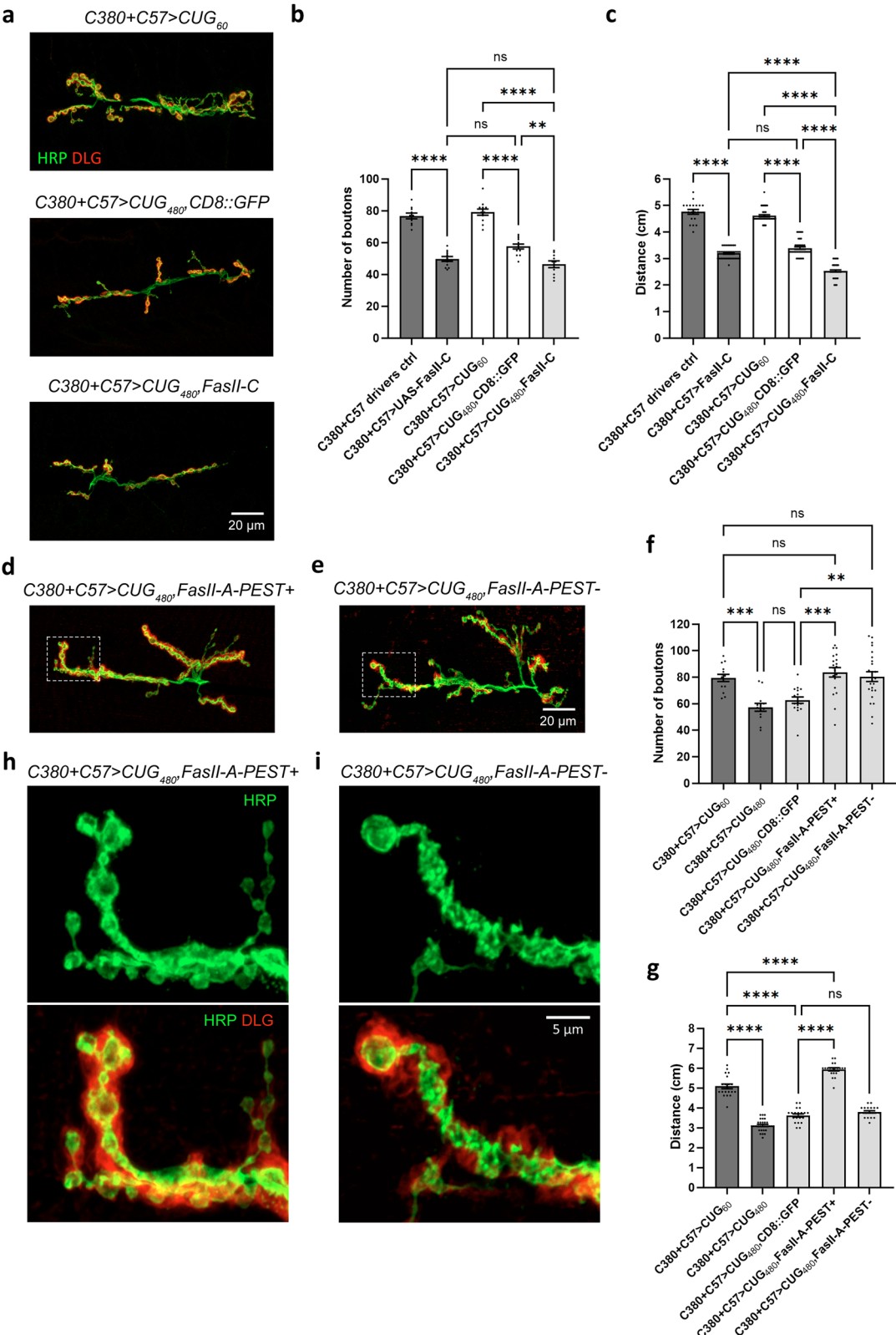

that $CUG_{480}$ expression induced upregulation of the two FasII-A isoforms, as well as FasII-C (and may have upregulated other uncharacterized FasII isoforms or dysregulated other proteins). In a previous study on the effects of FasII-A isoform overexpression[52], simultaneous pre- and postsynaptic overexpression of either FasII-A isoform led to an increase in QC[52]. QC was also increased when total-FasII was overexpressed[52]. These findings strongly indicate that the transmission phenotypes observed in $CUG_{480}$-expressing animals were likely to have resulted from the combined effects of several upregulated FasII isoforms, including FasII-A. Note that an increase in QC does not indicate increased muscle activity in the DM1 model. Rather, an increase in QC indicates an increase in the number of vesicles released per evoked response, which may be a homeostatic mechanism to compensate for impaired muscle output in $CUG_{480}$-expressing animals.

**Fig. 7 | Overexpression of FasII-C exacerbates and FasII-A rescues NMJ phenotypes of the *Drosophila* DM1 model. a, d, e** Confocal micrographs of *Drosophila* NMJs of late 3rd instar larvae at muscles 6 and 7 of segment A3. Anti-HRP (in green) marks the presynaptic boutons. Anti-DLG (in red) marks the postsynaptic density. Scale bar is 20 μm. **b** Quantification of bouton numbers in (**a**) and in other related genotypes. *n* = 11, 10, 11, 12, 12, 10, where *n* is the number of analyzed NMJs. **c** Quantification of larval locomotor activity. *n* = 40, 40, 40, 40, 40, where *n* indicates the number of analyzed larvae. **f** Quantification of bouton numbers in (**d, e**) and in other related genotypes. *n* = 14, 13, 17, 20, 24, where *n* is the number of analyzed NMJs. **g** Quantification of larval locomotor activity. *n* = 20, 30, 20, 20, 20, where *n* indicates the number of analyzed larvae. **h** High-magnification confocal micrographs of the white dotted rectangle in (**d**). **i** High-magnification confocal micrographs of the white dotted rectangle in (**e**). Each larva is defined as a biological replicate, and no more than two NMJs were analyzed per larva. One-way ANOVA with Tukey post-hoc test was performed. Histograms depict mean ± SEM. **p < 0.01, ***p < 0.001, ****p < 0.0001.

After observing that total *fasII* knockdown could rescue our DM1 model, we originally intended to investigate the effect of *fasII-C* knockdown using *UAS-FasII-C-RNAi*. This construct was generated by targeting exon 8 of *fasII*, which encodes the GPI anchor. However, we found that this construct also knocked down both *fasII-A* isoforms (Supplementary Fig. 3), which prompted us to rename it *UAS-FasII-A&C-RNAi*. We cannot exclude the possibility of a common off-target effect of RNAi that somehow allowed the amplicon to bind to other exons. A second possibility is that the two characterized FasII-A isoforms also each contain a GPI anchor in addition to a transmembrane domain. A third possibility is that other uncharacterized FasII isoforms contain a GPI anchor. A recent study indeed found a second GPI anchor-containing FasII isoform[53]. However, the reverse RT-PCR primer for *fasII-A* used in our study partially targeted the nucleotide sequence of exon 7 and the transmembrane domain. This suggests that the second GPI anchor isoform contains a transmembrane domain, in contrast to the findings of Neuert et al.[53]. Furthermore, when we used a pair of primers to exclusively amplify *fasII-A-PEST+* (the reverse primer targeted the PEST domain) (Supplementary Fig. 8b), we still observed *UAS-FasII-A&C-RNAi*-mediated knockdown (Supplementary Fig. 4), which strongly indicates that the RNAi line truly could knock down *fasII-A-PEST+*. The two latter possible explanations given above are not mutually exclusive and certainly warrant further investigation.

Both FasII and its mammalian orthologue NCAM1 belong to the immunoglobulin (Ig) domain superfamily and possess homophilic cell–cell adhesion mediator activity[54]. In humans, abnormal NCAM1-positive myofibers were observed in the deltoid muscles of patients with DM1[55], possibly due to increased expression of NCAM1. Furthermore, *NCAM1* transcripts were found to be upregulated in the frontal cortices of DM1 patients[56]. In agreement with these previous reports, our data showed increased NCAM1 protein expression in the spinal cord and tibialis anterior in DMSXL mice, as well as in the frontal cortices of patients with DM1 and transdifferentiated myotubes derived from patient cells. However, we observed that the deltoids of patients with DM1 expressed lower levels of NCAM1 than those of control individuals (Supplementary Fig. 12). In higher organisms, different types of skeletal muscles might exhibit different types of NCAM1 dysregulation due to expanded *CUG* repeat toxicity. Nevertheless, dysregulation of NCAM1 was still detected in human deltoids and is likely to have resulted in synaptic dysfunction at the NMJ. Alternatively, mature human muscle cells may have compensatory mechanisms that could explain the high level of NCAM1 in transdifferentiated myotubes but low level in deltoid muscles.

Our study results strongly suggest that both pre- and post-synaptically expressed *CUG* repeats participate to induce neuro-pathological phenotypes in presynaptic motorneurons. Thus, in the *Drosophila* larval NMJ, the pathological mechanism is likely to involve disrupted retrograde signaling, defined as communication from the postsynaptic muscle back to the presynaptic motorneuron. In *Drosophila*, the BMP signaling pathway is a major retrograde signaling pathway required for normal synaptic growth[57]. In this pathway, glass bottom boat (Gbb), a retrograde ligand molecule, is secreted by the muscle to activate presynaptic receptors[58]. At the presynaptic motorneuron terminal, Gbb is known to activate a receptor complex formed by wishful thinking (Wit) and thickveins (Tkv) to regulate synaptic growth[59]. Therefore, we attempted to rescue the DM1 model by expressing *UAS-Gbb-GFP* or *UAS-Tkv.CA* (Tkv.CA is a constitutively active form of Tkv receptor). Unfortunately, neither of these constructs rescued the decreased boutons on the DM1 background (Supplementary Fig. 13a, b). A specific level of Gbb signaling may be required for synapse integrity. Retrograde signaling molecules other than Gbb, such as Wnt, also may be involved.

Previous studies in mice have demonstrated that the *CUG* repeat-containing transcripts retained in the nucleus are recruited into ribonuclear foci and sequester RNA-binding proteins, such as MBNL1, thus compromising the RNA-splicing machinery[8,10]. In *Drosophila*, the molecular mechanism of *CUG*-induced toxicity is similar to that in the mouse model, involving the sequestration of muscleblind (Mbl) and other RNA-binding proteins by *CUG*-containing ribonuclear foci[26,27]. *mbl*$^{E27}$ is a homozygous lethal null mutation[27]. Thus, we analyzed heterozygous *mbl*$^{E27}$ mutants to determine whether they recapitulated the *CUG*$_{480}$ phenotypes. These mutants indeed exhibited reductions in the bouton and arbor disassembly phenotypes similar to those seen in *CUG*$_{480}$-expressing animals (Supplementary Fig. 14a–c). Furthermore, RT-PCR revealed that heterozygous *mbl*$^{E27}$ mutants exhibited upregulated expression of total *fasII* and *fasII-C*, similar to the patterns observed with *CUG*$_{480}$ overexpression (Supplementary Fig. 14d, e). These data suggest that the partial loss of Mbl functions in heterozygous *mbl*$^{E27}$ mutants might reflect the physiological conditions of the DM1 model. Our data also supported a role for Mbl in *fasII* processing and NMJ morphology in the context of DM1 neuropathology. Many other RNA-binding proteins also might be dysregulated by expanded *CUG* RNA, including Beag, Dsmu1, and embryonic lethal abnormal vision (Elav). Beag and Dsmu1 are spliceosomal proteins that were shown to participate in the pre-mRNA splicing of *fasII*[44]. Elav is a well-studied RNA-binding protein that regulates alternative splicing in neurons[60]. We performed semi-quantitative RT-PCR to examine the transcript levels of *beag*, *dsmu1*, *elav*, and *mbl* in *CUG*$_{480}$-expressing larval BWM. Our results showed upregulation of *beag* and *dsmu1*, indicating that these splicing factors are indeed dysregulated in the DM1 model (Supplementary Fig. 15). However, RED and DSMU1 (the mammalian homologs of Beag and Dsmu1) were not found to co-localize with *CUG* RNA foci in the nuclei of primary astrocytes from DMSXL mice (Supplementary Fig. 16). It is possible that the expanded *CUG* RNA led to the abnormal upregulation of *beag* and *dsmu1*, which contributed to part of the pathogenesis. It is also possible that the upregulation of Beag and Dsmu1 in *Drosophila* is a compensatory mechanism to counter the damaging effects of Mbl and FasII dysregulation.

As we have demonstrated that the expression of total NCAM1 is dysregulated in neural and muscular tissues from DMSXL mice (Fig. 3i–l), an important question concerns whether *Ncam1* RNA isoforms are also dysregulated in these animals. To answer this question, we subjected primary astrocytes and frontal cortex of DMSXL mice to RT-PCR analysis of distal exon selection. As expected, we indeed observed dysregulated usage of distal *Ncam1* terminal exons in these cells and tissues (Supplementary Fig. 17). In addition, we examined whether dysregulated *Ncam1* splicing could be caused by *Mbnl1/Mbnl2*

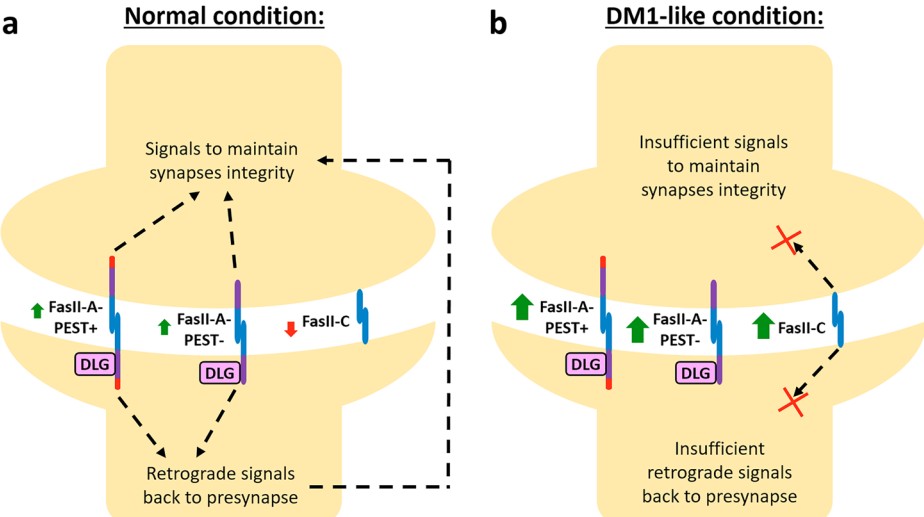

**a** Normal condition:

**b** DM1-like condition:

**Fig. 8 | Working model of synaptopathology in a *Drosophila* larval NMJ model of DM1. a** Under the normal condition, FasII-A-PEST+ and FasII-A-PEST− are the major isoforms present at the pre- and postsynaptic terminals. Both FasII isoforms convey intracellular signals to maintain synapse integrity. Postsynaptic FasII-A are capable of interacting with DLG, and may facilitate retrograde signals that help with maintaining bouton numbers. **b** Under the DM1-like condition, expanded *CUG* repeats causes upregulation of overall FasII, which in turn causes an abnormally high level of FasII-C at the synapse. FasII-C may bind to itself or compete with other isoforms to bind to the FasII-A isoforms. Since the FasII-C isoform lacks the cytoplasmic domain, its binding to FasII-A disrupts the intracellular signals required for maintaining synapse integrity.

double knockdown or knockout. Our results showed that *Ncam1* terminal exon splicing was indeed dysregulated in cultured astrocytes subjected to *Mbnl1/Mbnl2* double knockdown (Supplementary Fig. 18a, b) and in the frontal cortex of *Mbnl1/Mbnl2* conditional double-knockout mice (*Mbnl1*−/−; *Mbnl2*c/c; Nestin-Cre+)[61] (Supplementary Fig. 18c, d). As these defects parallel those observed in DMSXL mice (Supplementary Fig. 17), the shift in *Ncam1* terminal exon usage in DM1 is likely linked to functional depletion of MBNL proteins.

In our proposed model (Fig. 8), both FasII-A-PEST+ and FasII-A-PEST− are the major isoforms expressed at the presynaptic MNs and postsynaptic BWMs in a normal NMJ. These isoforms harbor cytoplasmic domains, allowing them to convey intracellular signals. At the postsynaptic BWM, the cytoplasmic PDZ-interaction motif of FasII-A isoforms interacts with DLG[48,49,54], which may facilitate retrograde signaling via molecules such as Gbb (*Drosophila* counterpart of mammalian BMP) to maintain bouton numbers[50]. Under the DM1-like condition, the expanded *CUG* repeats cause the upregulation of total FasII. Upregulated FasII-C may either bind to itself or compete with other isoforms to bind to FasII-A isoforms via their Ig domains (which mediate adhesion via transhomophilic binding). However, FasII-C does not contain a cytoplasmic domain and thus cannot transduce intracellular signaling like the FasII-A isoforms[54]. Hence, the binding of FasII-C to FasII-A disrupts the intracellular signaling required for proper synaptic functions, leading to defective synaptic transmission, locomotion impairment, and even gradual synapse disassembly.

The intricate mechanisms of synapse regulation involving NCAM1 in mammals are likely to be far more complex than those observed in *Drosophila*. Nevertheless, our study provides important foundational information about a basic mechanism of synapse dysregulation in a simple DM1 model. Specific levels of different mammalian NCAM1 isoforms may be expressed in different types of neurons and muscles to achieve the proper functions. The expansion of *CUG* RNA in DM1 probably disrupts the delicate regulation of NCAM1 and/or other cell adhesion molecules, leading to synaptic dysfunction. Although we cannot directly change the genetic make-up of a patient with DM1, we may be able to alleviate their CNS abnormalities by restoring the proper ratios of NCAM1 and/or other cell adhesion molecules at synapses.

## Methods

### Mouse tissue samples and ethics compliance

The DMSXL mice with an expansion of over 1300 CTG repeats and control mice used in the study were provided by Mário Gomes-Pereira and Geneviève Gourdon. For each genotype, 2 female and 3 male mice aged between 16 and 21 days were analyzed. All experiments involving the use of DMSXL research mice have complied with all relevant ethical regulations according to the Ministere de L'Enseignement Superieur, de la Recherche et de L'Innovation of France. Animal ethics was approved by Ministry of Higher Education, Research and Innovation (Paris). Authorization for animal experimentation number #23473. Animal facility approval number B751320. The study protocol was approved by Prefecture de Police (Paris) and the French Veterinary Department. Authorization for animal experimentation number 75003. Animal facility approval number B91228107. The *Mbnl1*−/−; *Mbnl2*c/c; Nestin-Cre+ conditional double-knockout mice (Mbnl1/Mbnl2 DKO) and wildtype (WT) P30 mouse brain samples were provided by Professor Maurice Swanson (Department of Molecular Genetics and Microbiology, Center for NeuroGenetics and the Genetics Institute, University of Florida, College of Medicine, USA). All Mbnl1/Mbnl2 DKO and WT animal procedures were approved by the Institutional Animal Care & Use Committee (IACUC) of the University of Florida (Approval number: IACUC202300000652).

### Human tissue samples and ethics compliance

All experiments using human samples were approved by the Ethics Committees of the host institutions. Written informed consent of specimen use for research was obtained from all patients. No payments of any form have been given to the subjects or their families. Human frontal cortex samples were collected from two different laboratories: Dr. Yasuhiro Suzuki (Asahikawa Medical Center, Japan) and Dr. Tohru Matsuura (Okayama University, Japan). Information relative to patients was previously described[15,62,63]. For the transdifferentiated myoblasts, samples were obtained from the Institute of Myology, Paris. In brief, skin biopsies were obtained from MyoBank-AFM bank of tissues for research, a partner in the EU network Euro-BioBank, in accordance with European recommendations and French legislation. Under these regulations, informed consent was obtained

from donors prior to biopsy collection. The samples originated from an 11-year-old female donor with DM1 (1300 CTG repeats) and a healthy 25-year-old male donor. The cells were immortalized using hTERT expression, which was previously described. Fibroblasts were transduced to express MyoD with the supplement of doxycycline, which drove the transdifferentiation of fibroblasts into myotubes[64]. For experiments involving deltoid muscles, muscle biopsies were taken after informed consent by patients and approval by the Experimentation Ethics Committee of the University Hospital La Fe (Valencia, Spain; authorization number: 2014/0799), according to the Helsinki Declaration.

## Fly Stocks
The following stocks were used: $w^{1118}$, $elav^{GS}$-GAL4 (43642), UAS-CD8::GFP (5137), UAS-Gbb-GFP (63057), and UAS-Tkv.CA (36537) were acquired from Bloomington Drosophila Stock Center. $UAS$-$CTG_{60}$ and $UAS$-$CTG_{480}$[27] were obtained from Ruben Artero. Transgene expression levels of these constructs can be found in (Supplementary Fig. 19). $UAS$-$CAG_{250}$[31] was obtained from Nancy Bonini. C380-GAL4 (a.k.a. BG380)[65] and C57-GAL4 (a.k.a. BG57)[65] were obtained from Vivian Budnik. Expression patterns of these drivers can be found in (Supplementary Fig. 20). UAS-FasII-A-PEST+[36,37], UAS-FasII-A-PEST−[66], UAS-FasII-C[66], UAS-Total-FasII-RNAi (a.k.a. UAS-Total-FasII-dsRNA or P[KK100888]VIE-260B[45], Vienna Drosophila RNAi Centre, #v103807), and UAS-FasII-A-RNAi[44] and UAS-FasII-A&C-RNAi (unpublished) were obtained from Brian McCabe. UAS-FasII-A&C-RNAi was originally named UAS-FasII-C-RNAi(#38) since it was designed against Exon 8 of fasII, which encodes for the GPI anchor of FasII-C. However, the construct was found to knock down FasII-A-PEST+ and FasII-A-PEST− as well (Supplementary Figs. 3 and 4). Hence, it was renamed as UAS-FasII-A&C-RNAi in this study. LexAop-FasII-A&C-RNAi was generated by subcloning the same RNAi construct (Amplicon sequence: GCTAATAACA ATCTCGGCAC GTTGCTCTAT TCGGCCGGAT TTAATTCCGG TGTCGGTGCG CTACACAAAC GACTGTTCAC AACAACAACA ACAACAACAG CCACATCAAC AACAACAATC ACATCGATAA CAACAGCAAC AACAACAATC ATTACGCTGG CCAC) into the pJFRC19-13XLexAop2-IVS-myr::GFP vector (Addgene plasmid #26224), and inserted onto the 2nd chromosome using the Phi3C1 system (The fly lines generation process was done by BestGene Inc., U.S.A.). nSyb-LexA (a.k.a. nSyb-LexA-GAD)[67] was obtained from Ching-Po Yang and Tzumin Lee. $Cass^M$-LexA (unpublished) was obtained from Vivian Budnik. Flies were reared in standard Drosophila medium at 25 °C. For genotypes involving $elav^{GS}$-GAL4, RU486 was dissolved in 100% ethanol before added to the medium to achieve a final concentration of 50 μM.

## Semi-quantitative RT-PCR
mRNA was extracted from either dissected larval CNSs or BWMs using standard TRIzol extraction method, homogenizing the tissues in TRIzol, adding chloroform to conduct phase separation, following with isopropanol RNA precipitation and resuspension of RNA using DEPC-treated water at the end[68]. Synthesis of cDNA was performed using the ImProm-II™ Reverse Transcription System (Promega). Primers used for amplification of fasII were as follow: Total-FasII Forward: GCAAC-CAGGTGGGATTAGG (on Exon 6), Total-FasII Reverse: TAACGCCCG-GACAGTATTTG (partially on Exon 6 and partially on Exon 7), FasII-A Forward: CTGTCCGGGCGTTAAGATC (partially on Exon 6 and partially on Exon 7), FasII-A Reverse: ACGTCAATTCCTCGTGTCG (partially on Exon 7 and partially on TM domain), FasII-A-PEST+ Forward: ACAC-GAGGAATTGACGTCATC (partially on Exon 7 and partially on TM domain), FasII-A-PEST+ Reverse: GTGGCTCCTTTACCAGCTG (on the PEST domain), FasII-C Forward: GCGTTAAGATCAGCGGCAC (on Exon 7), FasII-C Reverse: GAATCGGACTCACCTCGTG (partially on Exon 7 and partially on GPI anchor). The designs of these primers had been previously published[44]. Other primers used in this study were as follow: Beag Forward: GCATCAGAGGAGCCGATAATATC, Beag Reverse:

GGCCTTCTCGTTCTTCTCC. Dsmu1 Forward: TGCAGTTCTCGCGAGA-TAAC, Dsmu1 Reverse: TCACTGTAGCTGGCCAGTAG. Elav Forward: CAAGTCGCAGGTCTACATCG, Elav Reverse: CTCCTTTCGTCTGCGTA TCG. Mbl Forward: CAACGTGGAGGTCCAGAAC, Mbl Reverse: CCGGTCAGATAGGGGTTTG. Actin Forward: ATGTGCAAGGCCGG TTTCGC, Actin Reverse: CGACACGCAGCTCATTGTAG. SV40 Forward: GGAAAGTCCTTGGGGTCTTC, SV40 Reverse: GGAACTGATGAATGG-GAGCA. Hs promoter Forward: TCCTCCGAGCGGAGACTC, Hs promoter Reverse: TGGCAGATTTCAGTAGTTGCAG. All the primers were ordered and synthesized from the Thermo Fisher Scientific Inc. (Hong Kong). Quantification of gel bands was performed using ImageJ. The housekeeping gene, Actin, was used as a reference standard in each group to observe the expression level of the interested genes. Each of the interested gene band intensity was first normalized to its Actin band intensity. The fold change of each of the experiment group was then calculated by using the normalized intensity of each group over the normalized intensity of the control group.

## Semi-quantitative RT-PCR for the analysis of alternative exon usage
RNA extraction was performed with the RNeasy Mini kit (QIAGEN; 74104) following the manufacturer's protocol, including a DNase digestion step (RNase-Free DNase Set; QIAGEN; 79254) after the first wash with RW1 buffer. RNA concentration was assessed using the NanoDrop (Thermo Scientific) and RNA quality was verified by electrophoresis on agarose gel. cDNA synthesis and semi-quantitative reverse-transcriptase PCR analysis of alternative exon usage were performed by using PCR amplification (mostly around 21–26 cycles) to obtain PCR product following with resolved the PCR product through 2.5% (w/v) agarose gels and stained with ethidium bromide[62,63]. Oligonucleotide primers used for RT-PCR analysis of Ncam1 were the following: Ncam1 E16 (forward): ACGTCATGCTCAAGTCCCTG. Ncam1 E17 (reverse): AGTCACCGCAGAGAAAAGCA. Ncam1 E18 (reverse): ATGAGCAGGCCACACTTGTT.

## Locomotor behavioral assay for larvae
Larval crawling assay was previously published study[69]. In brief, wandering third instar larvae were loaded onto a 2% agarose plate with a grid (0.5 cm² per square) placed underneath. The total number of gridlines crossed by the animals in one minute was counted, and the actual distance calculated. n represents the number of larvae analyzed.

## Immunohistochemistry
Larval BWMs were dissected and fixed for 10 min in 4% paraformaldehyde, and permeabilized with 0.2% Triton-× 100 at room temperature[70]. After that, samples were incubated with primary antibodies overnight at 4 °C. Primary antibodies and their concentrations used: anti-HRP-Alexa Fluor 488 1:500 (Jackson), anti-HRP-Alex Fluor 647 1:500 (Jackson), anti-DLG1 1:750 (4F3, DSHB), anti-FasII 1:50 (affinity purified 1D4, DSHB)[71], anti-FasII 1:5 (34B3, DSHB)[42]. Secondary antibodies conjugated to Alexa Fluor 488/594 (Abcam) were used at a concentration of 1:200. Samples were incubated with secondary antibodies at room temperature for 1 to 2 h. Lastly, samples were mounted in the glass slice using anti-fade fluorescence mounting medium (Abcam, ab104135).

## Quantification of boutons
The number of type I boutons was obtained at muscles 6 and 7 of abdominal segment A3 of late 3rd instar larvae unless specified otherwise. n represents the number of analyzed NMJs. At most two NMJs were quantified in each animal, and each animal is defined as a biological replicate[70,72]. The development of larvae expressing $CUG_{480}$ using C380, $elav^{GS}$ (and fed with food with RU486) or other combinations involving these drivers were slower than that of controls by approximately 24 h. Thus, crosses involving these genotypes were set

up one day in advance so the dissections and immunostaining can be carried out at the same time. No notable abnormalities were found in the bouton numbers of the genetic controls for the major UAS lines used in this study (Supplementary Fig. 21).

## Quantification of arbor disassembly
An NMJ with at least one disassembling arbor was defined as an NMJ with arbor disassembly. A disassembling arbor was defined as a stretch of at least two boutons that are severed from the arbor and from one another (immunostained by anti-HRP), in which at least one of these boutons showed DLG staining on the postsynaptic muscle.

## Quantification of muscle areas
Lengths and widths of muscle 6 of segment A3 were measured under a 20× objective on a widefield microscope, and the areas were subsequently calculated (Supplementary Fig. 22). *n* represents the number of analyzed muscles. At most two muscles were quantified in each animal.

## Slot blot assay for *Drosophila* FasII and mice NCAM1
Fly heads, fly thorax muscles, mouse CNS tissues, or mouse muscle tissues were homogenized in the lysis buffer (100 mM Tris/HCl, pH6.8; 2% Sodium Dodecyl Sulfate; 40% w/v Glycerol). The homogenates were boiled at 99 °C for 10 min, then centrifuged at $14,000 \times g$ for 2 min and the supernatant was collected. Before the slot blot assay, samples were diluted with 2% SDS to a final volume of 200 µL and heated at 99 °C for 10 min. Protein samples were loaded to a 48-well Bio-Dot® microfiltration apparatus (Bio-Rad Laboratories, Hercules, CA, USA) with BioTrace™ NT nitrocellulose membrane (Pall Life Sciences, Portsmouth, UK; pore size 0.2 µm). The membrane was blocked at room temperature in 5% non-fat milk and incubated in 34B3 (Developmental Studies Hybridoma Bank, Iowa City, IA, USA; Mouse; 1:10 https://dshb.biology.uiowa.edu/)[42] at 4 °C overnight for the detection of FasII in fly protein samples. For mouse protein samples, the membrane was incubated in ab154566 (Abcam, Cambridge, UK; Rabbit; 1:1000) for NCAM1 detection. The protein chemiluminescence signal was obtained and visualized with the ChemiDoc™ Touch Gel Imaging System (Bio-Rad Laboratories, Hercules, CA, USA). After FasII or NCAM1 detection, the membrane was stripped in stripping buffer (Thermo Fisher Scientific, Grand Island, NY, USA) and blocked again in 5% non-fat milk. β-tubulin was used as the internal loading control, where the membrane was re-probed with ab6046 (Abcam, Cambridge, UK; Rabbit; 1:500) for fly protein samples or 2G7D4 (GenScript, Piscataway, NJ, USA; Mouse; 1:2000) for mouse protein samples, respectively. The images were analyzed with ImageLab™ software (Bio-Rad Laboratories, Hercules, CA, USA) and band intensities were quantified using the Image J software (Research Services Branch, National Institute of Mental Health).

## Slot blot assay for human frontal cortex
Total protein was extracted from 20 to 30 mg human frontal cortex using RIPA buffer (Pierce™ RIPA Buffer, ThermoScientific, 89901) supplemented with 0.05% CHAPS (Sigma, C3023), 1× complete protease inhibitor (Sigma-Aldrich, 04693124001), 1× PhosSTOP phosphatase inhibitor (Sigma-Aldrich, 04906845001), and 1 mM sodium orthovanadate (Sigma, S6508). Protein concentrations were determined using the Pierce BCA Protein Assay Kit (Thermo Scientific, 23227). Porablot NCP nitrocellulose membranes (Macherey-Nagel, 741280) and Bio-Dot® SF filter paper (Bio-Rad, 1620161) were soaked and equilibrated in 1× PBS (Gibco, 20012-019) and then placed in the Bio-Dot® SF Apparatus (Bio-Rad, 1706542). Membrane slots were rinsed with 100 µL 1× PBS, and then loaded with 20 µg frontal cortex protein in 100 µL RIPA buffer under vacuum. The membranes were carefully removed from the Bio-Dot® SF Apparatus and stained with ATX Ponceau S red solution (Sigma-Aldrich, 09189-1L-F) for total

protein quantification. Membranes were blocked in 5% Blotto (Santa Cruz Biotech, sc-2324) diluted in 1× TBS-T (10 mM Tris-HCl, 0.15 M NaCl, 0.05% Tween 20) for 1 h at room temperature, and incubated overnight at 4 °C with primary anti-NCAM1 antibody (GeneTex, GTX111684) diluted 1:10,000 in 5% Blotto. After three washes in 1× TBS-T, membranes were incubated with secondary HRP-conjugated goat anti-rabbit IgG (Life Technologies, 31460) diluted 1:5000 in 5% Blotto, for 1 h at room temperature. Membranes were finally washed another three times in 1× TBS-T and developed with Clarity Max ECL Substrate (BioRad, 1705062). Image acquisition and quantification were conducted with BioRad ChemiDoc™ MP Imaging System and software (Image Lab 6.0.1). Relative protein levels in non-DM controls were normalized to 1 and statistical analysis was performed with Prism (GraphPad Software, Inc.).

## Transdifferentiation of myotubes
Control or DM1 patient myoblasts were differentiated for 7 days into transdifferentiated myotubes. Briefly, the cells were first immortalized using hTERT expression. After that, fibroblasts were transduced to express MyoD in the presence of doxycycline. This process induced expression of MyoD facilitates the transdifferentiation of fibroblasts into myotubes[64].

## Quantitative dot blot assay
1 µg/well of protein samples were denatured (100 °C for 5 min) and loaded in quantitative dot blot (QDB) plates (Quanticision Diagnostics Inc, Research Triangle Park, NC, USA). Each cell sample was loaded in quadruplicate in two different plates; one was used to detect NCAM11 (1:1000, Proteintech, Manchester, UK) and the other for GAPDH (1:500, (G-9) Santa Cruz Biotechnology, Dallas, TX, USA), which was used as an endogenous control. ECL substrate (Pierce™ ECL Western blotting substrate) was used to detect and visualize the result and TECAN infinite M200 Pro with an initial acquisition time of 1 s/well in the luminescence was used to acquire data[73].

## Electrophysiology
Electrophysiology was performed as previously published study[74]. For detailed methods on how to perform Drosophila NMJ electrophysiology, please see published protocols[75,76]. Briefly, wandering third instar larvae were collected and filleted for NMJ analysis. Driver control and experimental electrophysiological recordings were performed in parallel using identical conditions. Larval dissections and recordings were performed in a modified, low-magnesium HL3 saline[77]. (70 mM NaCl, 5 mM KCl, 5 mM HEPES, 10 mM NaHCO$_3$, 115 mM sucrose, 4.2 mM trehalose, 0.5 mM CaCl$_2$ (unless otherwise noted), 10 mM MgCl$_2$, pH 7.2. Neuromuscular junction sharp electrode recordings were performed at muscles 6 and 7 of abdominal segments A2 or A3. For a muscle to be acceptable for recording, it needed to have an input resistance of $\geq 4 \, M\Omega$ and a resting potential more hyperpolarized than $-58 \, mV$.

Recordings were performed on an Olympus BX51WI microscope and acquired using an Axoclamp 900 A amplifier, Digidata 1440 A acquisition system, and pClamp10.7 (Molecular Devices) software. Data were analyzed using MiniAnalysis (Synaptosoft) and the Clampfit (Molecular Devices) programs. mEPSPs and excitatory postsynaptic potentials (EPSPs at 1 Hz stimulus) were collected[74]. For each NMJ recorded, four quantifications were made: quantal size (mEPSP size); quantal frequency (mEPSP frequency); evoked potential size (EPSP size); and quantal content (QC). In the case of QC, uncorrected QC was calculated per NMJ by dividing the average EPSP by the average mEPSP.

## Fluorescent in situ hybridization (FISH) and immunofluorescence
Primary astrocytes were cultured for 2 weeks, washed 3 times in 1× PBS and then fixed for 15 min in 4% PFA (J61899.AP, VWR) at 4 °C. Following

5 × 2 min washes in 1× PBS, cells were permeabilized in 2% ice-cold acetone in 1× PBS for 5 min at room temperature. Prehybridization was carried out in 30% formamide/2× SSC for 10 min at room temperature. Hybridization with 1 ng/mL 5′-Cy3-labeled (CAG)5 PNA probe in hybridization buffer (30% formamide, 2× SSC, 0.02 %BSA, 66 mg/ml yeast tRNA, 2 mM vanadyl complex) was carried out for 2 h at 37 °C in a dark humidified chamber. Next, cells were washed for 30 min in 30% formamide/2× SSC at 50 °C, followed by one last wash of an additional 30 min in 1× SSC at room temperature.

Cells were then incubated in blocking and permeabilization solution for 1 h at room temperature (1× PBS; 0.1% Triton ×-100; 10% normal goat serum, G6767, Sigma). Primary antibody was diluted in permeabilization (1× PBS; 0.1% Triton ×-100) and incubated over night at 4 °C (SMU1, Sc-100896, Santa Cruz biotechnologies, dilution 1/500; RED, PA5-101789, Thermo Fisher, 1/500). Next day, following 3 × 10 min washes in 1× PBS, cells were incubated with secondary antibody diluted in permeabilization solution for 1 h at room temperature. Excess antibody was washed 3 × 10 min in 1× PBS, and incubated with 0.0002% DAPI (10236276001, Sigma) for 15 min at room temperature. Cells were finally washed 3 × 10 min in 1× PBS and mounted with Dako Fluorescent Mounting Medium (S302380, Dako) prior to observation. Images were acquired with a laser apotome (Zeiss, Axio Observer), using a 40× objective and Zeiss Zen software.

### Statistical analysis
For comparisons between three or more sample groups, an analysis of variance (one-way ANOVA) with Tukey post-hoc test was performed. For pair-wise comparisons, Student's $t$-test (two-tailed) was used. $*P < 0.05$, $** P < 0.01$, $*** P < 0.001$, $**** P < 0.0001$. All histograms depict mean ± S.E.M. All experiments were performed at least three times independently.

### Ethics
Biological and chemical safety approval (14102220) from The Chinese University of Hong Kong was obtained for this study. All animal procedures were approved by the CUHK Animal Experimentation Ethics Committee (and their care was in accordance with the institutional guidelines).

### Reporting summary
Further information on research design is available in the Nature Portfolio Reporting Summary linked to this article.

## Data availability
All data supporting the findings of this study are presented within the main manuscript and Supplementary Information. All the data generated in this study are provided in the Supplementary Information/ Source Data file. Source data are provided with this paper.

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

## Acknowledgements

This work was supported by Research Grants Council of Hong Kong (14102220), Jérôme Lejeune Foundation (GRT-2022A/2120), Agence Nationale de la Recherche (ANR, France; Project ANR-22-CE12-0028-01), and Association de l'Institut de Myologie (AIM, France; Project Grant "Atteinte neurologique"). We would also like to thank Professor Maurice Swanson (Department of Molecular Genetics and Microbiology, Center for NeuroGenetics and the Genetics Institute, University of Florida, College of Medicine, USA) for sharing of the *Mbnl1*[-/-]; *Mbnl2*[e/e]; Nestin-Cre[+] P30 mouse brain samples and Dr. Ching-Po Yang and Professor Tzumin Lee (Life Sciences Institute, University of Michigan, USA) for sharing of *Drosophila* reagents. Any opinions, findings, conclusions, or recommendations expressed in this event/publication (or by members of this State Key Laboratory) do not reflect the views of the Government of the Hong Kong Special Administrative Region or the Innovation and Technology Commission.

## Author contributions

A.C.K. and K.Y.W.Y. designed and performed most experiments; Y.W., L.I.L., J.T.P.C., Z.S.C., S.I.P., and J.M.S.F. contributed to various experiments, including RT-PCRs, slot blot analyses of FasII, dissections for NMJ analyses, and larval behavioral assays; N.S.A. and C.A.F. contributed to electrophysiology experiments and analyses; A.B., N.M., J.P., J.V., and R.A. provided human DM1 patient samples and patient cell-derived samples, performed quantitative dot blot analyses, and provided numerous key fly lines for the study; P.M. and M.G.-P. provided human DM1 patient frontal cortex samples, performed slot blot analyses and splicing analysis of NCAM1 and immunohistochemistry of RED and SMU1; A.H. and G.G. provided DMSXL mouse samples and contributed to initial RT-PCR and Western blot experiments of NCAM1 analysis; C.K.B., M.Z., V.B., and T.T. generated the *Cass*[M]-*LexA* fly line, provided several antibodies for the study, and performed the dissections and NMJ analyzes; E.S.B. and B.M. generated the *UAS-FasII-A&C-RNAi* fly line and provided numerous key fly lines for the study; A.C.K., K.Y.W.Y., L.I.L., A.B., M.G.-P., C.A.F, G.G., and H.Y.E.C. contributed to manuscript writing; H.Y.E.C. supervised and directed the project.

## Competing interests

The authors declare no competing interests.

## Additional information

[1]School of Life Sciences, Faculty of Science, The Chinese University of Hong Kong, Shatin N.T, Hong Kong SAR, China. [2]Gerald Choa Neuroscience Institute, The Chinese University of Hong Kong, Shatin N.T, Hong Kong SAR, China. [3]Interdisciplinary Graduate Program in Neuroscience, Department of Anatomy and Cell Biology, Carver College of Medicine, University of Iowa, Iowa City, IA, USA. [4]Sorbonne Université, Inserm, Institut de Myologie, Centre de Recherche en Myologie, Paris, France. [5]CIBER de Enfermedades Raras, Instituto de Salud Carlos III, Madrid, Spain. [6]Incliva Biomedical Research Institute, Valencia, Spain. [7]Human Translational Genomics Group, University Institute of Biotechnology and Biomedicine (BIOTECMED), University of Valencia, Burjassot, Spain. [8]Neuromuscular and Ataxias Research Group, Health Research Institute Hospital, La Fe (IIS La Fe), Valencia, Spain. [9]Department of Neurobiology, University of Massachusetts Chan Medical School, Worcester, MA, USA. [10]Department of Neurology, Icahn School of Medicine at Mount Sinai, New York, NY, USA. [11]Brain Mind Institute, EPFL - Swiss Federal Institute of Technology, Lausanne, Switzerland. [12]State Key Laboratory of Agrobiotechnology (CUHK), The Chinese University of Hong Kong, Shatin N.T, Hong Kong SAR, China. [13]These authors contributed equally: Alex Chun Koon, Ka Yee Winnie Yeung. ✉e-mail: hyechan@cuhk.edu.hk

