## [Transparent Peer Review file · Nature Communications]

Pre- and postsynaptic upregulation of FasII synergistically underlies neuropathological and behavioral phenotypes in a *Drosophila* model of myotonic dystrophy

Corresponding Author: Professor Ho Yin Edwin Chan

Version 0:

Reviewer comments:

Reviewer #1

(Remarks to the Author)

Despite decades of research, the neuropathological mechanisms underlying Myotonic dystrophy type 1 (DM1) are still not fully understood. In this study, the authors model DM1 neuropathology by expressing untranslated expanded CUG repeats at the neuromuscular junction in *Drosophila* larvae. They observed that both presynaptic (motoneuron) and postsynaptic (muscle) expression of CUG repeats led to a reduction in synaptic boutons, increased arbor disassembly, and impaired larval locomotion. The reported findings further indicate that CUG repeat expression elevated levels of the cell adhesion molecule FasII (analogous to NCAM1 in mammals) in both motor neurons and body wall muscles. They report that reducing fasII expression did restore bouton numbers and improved locomotor activity. Further analysis suggested that the upregulation of the FasII-C isoform was primarily responsible for these phenotypic changes. Additionally, overexpressing the FasII-A-PEST+ isoform helped mitigate synaptic and behavioral abnormalities, likely by outcompeting the excess FasII-C. Through their research, they provided insights into synapse dysregulation in DM1, offering a foundational understanding of the molecular mechanisms involved in this disorder.

In this manuscript, Koon et al. explored the effects of expressing CUG repeats in the developing larval neuromuscular junction (NMJ), demonstrating some neuropathological aspects of a previously well-established *Drosophila* model of myotonic dystrophy type 1 (DM1). The authors mainly used NMJ bouton number and larval locomotor activity as readouts, and dissected the effects of pre- and/or postsynaptic CUG repeats expression. They show that either pre- or postsynaptic CUG repeats expression resulted in locomotor deficits, without affecting the gross NMJ structure when analyzed as NMJ bouton numbers. Furthermore, simultaneous pre- and postsynaptic CUG repeats expression triggered both locomotor and NMJ structural defects. They then show that increased FasII level might contribute to these structural and behavioral defects. While the link between the *Drosophila* DM1 model and FasII might per se interesting, this study has notable conceptual and methodological shortcomings. The argument for specific distinct molecular mechanisms connecting DM1 pathology to FasII isoforms lacks convincing evidence. The authors appear to over-interpret their data, particularly regarding the claimed "outcompetition of FasII-C by FasII-A," which does not seem convincingly supported by their findings. Additionally, it is surprising that the authors opted to test only FasII-C overexpression in electrophysiological experiments, leaving the DM1 model unexamined in this context. This choice significantly limits the study's comprehensiveness and the robustness of its conclusions. To strengthen their findings, the authors should include a broader range of electrophysiological tests in the DM1 model and provide clearer evidence to support their claims about the molecular mechanisms involving FasII isoforms. Further major concerns:

1. The authors chose to utilize the more accessible larval neuromuscular junction (NMJ) to investigate the neuropathology of DM1. However, they did not incorporate any functional readouts, such as electrophysiology, to assess potential changes in neurotransmission associated with DM1. Instead, they focused solely on the electrophysiological phenotypes of FasII-C overexpression (OE), which cannot be considered a valid proxy for DM1. This approach significantly limits the study's ability to provide comprehensive insights into DM1 neuropathology and its impact on neurotransmission.
2. The authors did not investigate potential signaling pathways known to mediate NMJ bouton structure, such as BMP and Wnt signaling. There are numerous established tools available in the field for studying these pathways, and the authors should utilize these to elucidate the mechanisms regulating NMJ structural phenotypes in DM1. Currently, the connection between DM1 and FasII does not sufficiently explain the observed NMJ structural phenotypes. By exploring these signaling pathways, the authors could provide a more comprehensive understanding of the molecular mechanisms underlying the structural abnormalities seen in DM1.
3. The authors proposed the hypothesis of "FasII isoform competition to bind to isoform A" based on the distinct phenotypes observed in NMJ boutons and larval locomotion after FasII isoform knockdown or overexpression. This hypothesis appears

to be speculative and lacks robust support from the provided data and existing literature. Additionally, the potential differing strengths of the C380 and C57 Gal4 lines and fasII RNAi lines, as shown in Figure S1b, may further complicate the situation.

4. Using the number of NMJ boutons as the primary and nearly exclusive indicator of synaptic abnormalities is superficial, also in the context of DM1. For instance, in Figure 4, although FasII-A-PEST+ overexpression does not affect NMJ bouton numbers, it does result in a gain-of-function in locomotion. This discrepancy suggests that the NMJ bouton count alone may not fully capture the synaptic changes in DM1. Furthermore, while the authors mention the use of anti-FasII (1D4) in immunostaining methods, it is surprising that they did not perform FasII staining using isoform-specific antibodies like 1D4 versus 34B3. Conducting NMJ immunostaining against FasII would provide critical spatial information regarding FasII distribution in DM1 and FasII overexpression models. Such information would be essential for understanding the precise role of FasII isoforms in NMJ structural phenotypes and could significantly enhance the mechanistic insights derived from the study.

Minor points:

1. In the abstract the authors state, "In this study, we characterized a novel model of DM1 neuropathology by expressing untranslated expanded CUG repeats at the Drosophila larval neuromuscular junction." This in my eyes is misleading, as the use of CUG repeat expression to model DM1 in Drosophila was proposed over 15 years ago. Thus, the model itself is not that novel. Am I right here? If yes the authors should clarify this point to avoid overstating the novelty of their approach and to accurately reflect the existing body of research.
2. In Figure 2, was elav-GeneSwitch/C57-driven DM1 model also sufficient to cause NMJ bouton phenotypes like with C380/C57 ?
3. Quantification is missing in Figure 3a,b.
4. In Figure 3c,d, the NMJ bouton phenotypes and larval locomotion of fasII-A and A&C RNAi are shown only in DM1 background, the effects of fasII-A and A&C RNAi alone are missing
5. In Figure 4b, CUG-60 expression is not the correct control for FasII OE
6. Similar in Figure 4d,e, CUG-60 expression is not the correct control for FasII-C OE
7. In Figure 5, the authors differentiated pre- and post- OE of FasII-C in electrophysiology. However, the authors did do the same in Figure 4.
8. In Figure 6, did the authors also performed fasII-C OE in DM1? Were there additive effects in NMJ bouton number and locomotion, and electrophysiology
9. Except for DM1 model flies, genetic controls for all the UAS lines including RNAi lines and cDNA OE line are missing. Given that genetic background affects NMJ bouton phenotypes, it will be important to include those UAS controls.
10. It is also a bit surprising that, given the per se limited electrophysiological data of Figure 5, the authors did not show any sample traces for minis and evoked.

Reviewer #2

(Remarks to the Author)

Kun et al., report CUG repeat dependent upregulation of FasII in Drosophila model of DM1 showing that it causes neuromuscular junction (NMJ) defects and affected mobility. Interestingly, FasII ortholog NCAM1 expression is also upregulated in muscular and neural tissues of DM1 mouse model (DMXL) and in DM1 patients. Authors take advantage of the well characterised Drosophila NMJ system in 3rd instar larva to test influence of toxic CUG repeats on presynaptic (motor neurons) and post-synaptic (body wall muscles) components of NMJ. They apply genetic tools available in fly model to rescue observed DM1-associated NMJ phenotypes and larva mobility. Authors propose a model in which the expression of FasII isoforms is affected in DM1 context with aberrant upregulation of FasII C isoform that impacts synapse integrity. The originality of the manuscript resides in assessing effects of toxic repeats in pre- and post-synaptic components of the NMJ and how they impact the neuromuscular synapse.

This experimental setup allows identification of FasII/NCAM1 gene deregulations conserved between fly and human which could inspire future investigations in the field.

The manuscript suffers however from some inconsistencies. It would also benefit from applying image analyses tools to better describe NMJ phenotypes and from linking at molecular levels toxic repeats expression with the deregulation of FasII. Inconsistencies :

1. Authors refer in the title and in the text to neuropathological and behavioral defects in DM1 but they analyse phenotypes of neuromuscular junctions and larval mobility. Whether dysregulation of FasII influences fly behavior and neurological CNS functions is not shown.
2. On page 4 authors say that they overexpressed control UAS-CUG60 construct (which should rather be UAS-CTG60) with the number of CTG repeats that is lower than the disease threshold. This is not correct as the number of CTG repeats in dmpk gene higher than 37 is already classified as potentially DM1 pathogenic.
3. On page 6 authors comment that « Previous studies suggested that the expression of CUG480 in the postsynaptic BWMS alone was sufficient to produce locomotor defects in Drosophila larvae ». They refer to Picchio et al., 2013. This is not correct. In this paper larval locomotion was analysed using inducible UAS-240, UAS-600 and UAS-960 CTG repeat lines but not UAS-480. Also in Picchio et al., repeats expression was driven by Mef-GAL4 showing much earlier muscle expression than C57-GAL4 used by authors.

General comment :

Authors hypothesize that FasII could be dysregulated by CUG repeat transcripts, Underlying question is whether this dysregulation is dependent on Mbnl1, major RNA-binding protein sequestered by toxic CUG repeats.

Muscle targeted knockdown of Muscleblind (Mbl), Drosophila ortholog of Mbnl1, mimics partial loss of Mbnl1 and could be

used to test whether NMJ phenotypes and FasII deregulation is Mbl-dependent.

Specific comments:

Fig. 1

- Fig.1a - In the presented view the reduction of button number in CUG480 versus CUG60 is not evident to see....A larger view showing NMJs of muscle 6 and 7 would help to appreciate the reduction in button numbers and the retraction of arbors.
- It is surprising that an increase in arbor disassembly observed in C380-driven presynaptic overexpression of CUG480 does not reduce the button number.
- Criteria for quantification of arbors disassembly are not clear. Segmentation and image analyses tools allowing to measure arbor's surface/extensions need to be applied to assess NMJ arborisations in different genetic contexts.
- Fig. 1d – Dlg signal is saturated. A larger view of NMJs is required.
- It would be appropriate to present expression pattern of the two drivers (C57 and C380) used in this study (in a supplementary data).

Fig. 2

Considering that several FasII isoforms are produced and differentially dysregulated in DM1 context, western blot would allow to visualize isoform specific dysregulation. Using slot blot and not western blot when assessing FasII protein expression in *Drosophila* is surprising.
FasII ortholog NCAM1 in DM1 mouse models and in DM1 patients is also upregulated. Is NCAM1 presenting several isoforms that are differentially dysregulated in DM1?

Fig. 3

This figure would benefit from a scheme of FasII gene with different FasII isoforms. According to Flybase, FasII encodes 7 different isoforms. Why authors focus on FasII A and FasII C only? If alternative splicing is in play, how FasII splice variants are expressed in DM1 contexts.

Fig.4 Simultaneous overexpression of FasII C in MNs and BWMs leads to arbors disassembly.
Did authors check effects of FasII C overexpression in MNs only?

Fig.6 – Link between buttons number and locomotor capacities is unclear. Why increase in buttons number in FasII A-PEST-rescue context does not result in mobility improvement?

Reviewer #3

(Remarks to the Author)

This is a very nice paper with interesting phenotypes and genetics. In this study, the authors reveal the function of fasII in disassembly and impairment of locomotor activity in the *Drosophila* DM1 model. While the manuscript is well-written and likely to contribute to the field, it could be further improved with additional data and analyses, which are crucial for strengthening the conclusions.

1) The connection to DM1 is tenuous. NCAM1 changes in DM1 seem variable depending on what tissue or dataset they are looking at. Also, there is no clarity to what these changes mean. Similar things can be said about the data from DMSXL mice. The data doesn't say anything about where the NCAM is being expressed (what cells, etc) and so has no relevance in the context of this manuscript.

2) There is no data on the expression level of the CUG transgene in the various conditions. This is essential to ensure that this element of the experiment is controlled and assessed.

3) Do the CUG repeats get expressed in the NMJ and which cells if any have RNA foci? For example, in Figure 1, although it has been shown previously that CUG480 RNA is expressed in this model, fluorescence in situ hybridization (FISH) analysis will help show that CUG480 RNA is also expressed in the pre-synaptic MNs and post-synaptic BWMs. Does Mbl get sequestered in those cells? Is Mbl expressed in the pre-synaptic and post-synaptic cells? Does the RNA get expressed there?

4) Does an Mbl knockout recapitulate the CUG480 phenotype? Does it or does it not? Answer would be interesting either way.

5) There is no description of how the quantification of FasII RNA levels were carried out. Was it slot blot or quantitative real-time RT-PCR. The first method is inadequate. This study's findings depend on the upregulation of fasII (cell adhesion molecule) in different models. If done quantitatively, the upregulation of fasII data could be more convincing (Figure 2). Authors need to verify the upregulation of fasII by different methodologies, too.

6) There is an implication in this study that there is an imbalance in the isoforms of FasII. No clear studies were done to assess the relative levels of the isoforms at both the levels of the RNA and protein. This would be essential.

7) FasII protein expression at the NMJ/boutons should have been done using isoform specific and total FasII antibodies, as described in reference 39. For example, Figure 3 shows the rescues of NMJs in the Drosophila DM1 model, but no data shows how much knockdown was achieved.

8) Similarly, western blots should have been done to assess the relative protein levels of the various isoforms.

9) The studies done by Beck et al (ref 39), identifying beag and dsmu1 mutants and the essential role of these proteins in splicing of FasII, are highly relevant to this current study. They aren't even mentioned, other than a passing reference to the fact that the flies used in that paper were used in this study. The phenotypes mentioned in that paper are highly reminiscent of the results from the current study. This includes the importance of the FasII-A-PEST+ isoform for regulation of NMJ growth and bouton number. This can't be ignored. The authors are trying to make a connection to splicing defects in DM1, but ignoring the splicing factors that have already been reported to be relevant in DM1 and in FasII splicing.

10) Are there antibodies for Beag? If so, what are the protein levels of Mbl, Elav (the fly homolog of ELAV proteins in mammals), Beag and dsmu1 in the current experiments? Does Beag co-localize with the RNA foci and Mbl? Does FLAG-tagged Beag (see Ref 39) co localize with the RNA foci or Mbl? What are the expression levels of Beag and dsmu1 in the CUG480 model, at the RNA and protein level? The mammalian homolog of Beag is called RED (if I am correct) and there are antibodies for that.

There are a couple of grammatical errors in the abstract section and elsewhere.

Reviewer #4

(Remarks to the Author)

I co-reviewed this manuscript with one of the reviewers who provided the listed reports as part of the Nature Communications initiative to facilitate training in peer review and appropriate recognition for co-reviewers.

Version 1:

Reviewer comments:

Reviewer #1

(Remarks to the Author)

I appreciate the considerable effort the authors have put into revising their manuscript. It is clear that they took the comments seriously and have added a number of new experiments, including electrophysiological analyses, immunostainings, and initial pathway tests. The presentation has also improved, and the revised version reads more clearly than before. I fully acknowledge and value this commitment and the amount of work that went into the revision.

That said, I must admit that I still remain with my main reservations. My initial concerns were not about a single missing experiment, but rather about the overall mechanistic depth of the study. While the new data extend the descriptive part of the work, they do not substantially strengthen the causal or mechanistic link between CUG-repeat toxicity and FasII dysregulation. The proposed pre- and postsynaptic "synergistic" effects also remain interpretative rather than experimentally dissected. In this sense, the revision does not really change the conceptual level of the study.

From my own perspective as someone working on synaptic mechanisms, the study remains primarily correlative and lacks the mechanistic resolution that would turn it into a conceptual advance. It is also a little difficult for me to judge the specific relevance for the DM1 field. If my expert colleagues in that area felt that the work represents a clear and important step forward, I would not object to publication. However, from the viewpoint of synaptic biology, I am sorry to say that I do not see the added value or conceptual advance that would justify publication in Nature Communications.

Reviewer #2

(Remarks to the Author)

Authors in response to my and other reviewer's comments improved significantly their manuscript. One of important supplementary data provided in the revised version is the analysis of a reduced Mbl level. The authors show that the heterozygous Mbl mutant context mimics toxic CTG repeats and FasII OE effects on NMJ. This finding reinforces authors conclusions. However, when using heterozygous Mbl mutant authors do not address specific pre- versus post-synaptic roles of Mbl attenuation (analysed in the contexts of CTG repeats and Fas2). It would be thus appropriate and of interest to the field to provide this analysis (using UAS-MblRNAi line(s)).

Regarding author's response to my first comment and to be more precise: As the DM1 neuropathology includes brain impairment authors need to clarify which aspects of neuropathology they analyse.

One possibility could be by modifying the last sentence in the introduction.

For example:

"These findings provide important new insights into the mechanisms underlying motor defects associated with DM1 neuropathology".

Reviewer #3

(Remarks to the Author)

The authors have made a genuine effort to address the queries of all three reviews. The manuscript is all the better for it.

I still have a question that remains unanswered. The authors should clarify if they think that NCAM1 expression is upregulated in DMSXL and in DM1.

The other thing is a technical issue. I am not sure that the way the assay seems to be described for supplemental figures 17 and 18 is the correct way to assess alternatively spliced isoforms of NCAM. It seems like they used primers situated in exon 16 and 17 for the isoform that includes exon 17 and a primer pair located in exon 16 and 18 for the other. But, if exon 17 is alternatively spliced, two products should have shown up in the exon 16-18 product; one representing exons 16,17,18 and the other being exons 16 and 18 (i.e. exon 17 is spliced out). But that is not what the data shows. The way the results appear, it looks like there is no alternative splicing of exon 17. The Sashimi plot confirms that. If you look at it, it looks like there are transcripts that contain exon 16 and 17 that seem to terminate there (i.e. no 3' exons are joined to this transcript), and there are no transcripts that contain exon 16, 17 and 18. This is not an alternative splicing pattern.

If the implication of the data is that MBNL proteins are causing alternative splicing defects in NCAM in the mouse situation, as the MBNL proteins do for 100s of other transcripts in DM1, and that is how we tie the effects on Fas in flies to the spliceopathy model of RNA toxicity in DM1, then the conclusion is incorrect.

Please correct your discussion and don't imply splicing defects in NCAM for the human or mouse situation. Your data doesn't show it.

It is more likely a transcriptional upregulation.

Otherwise, you have done a commendable job and I have no other issues.

POINT-BY-POINT RESPONSE TO REVIEWER COMMENTS

Dear Reviewers,

For your convenience, we have included low resolution versions of the main figures (and the figure legends) in the manuscript where the figures are being cited. But for the high resolution versions of the figures, please kindly refer to the attached figures at the end of the manuscript. Thank you for your attention.

Reviewer #1 (Remarks to the Author):

While the link between the *Drosophila* DM1 model and FasII might per se interesting, this study has notable conceptual and methodological shortcomings. The argument for specific distinct molecular mechanisms connecting DM1 pathology to FasII isoforms lacks convincing evidence. The authors appear to over-interpret their data, particularly regarding the claimed “outcompetition of FasII-C by FasII-A,” which does not seem convincingly supported by their findings.

Response: We thank the reviewer for pointing this out, and we apologize for over-interpreting this data. Since we don't have additional support for this hypothesis, we have decided to remove this interpretation from the Results section but to introduce this possible hypothesis in the Discussion section.

Additionally, it is surprising that the authors opted to test only FasII-C overexpression in electrophysiological experiments, leaving the DM1 model unexamined in this context. This choice significantly limits the study's comprehensiveness and the robustness of its conclusions. To strengthen their findings, the authors should include a broader range of electrophysiological tests in the DM1 model and provide clearer evidence to support their claims about the molecular mechanisms involving FasII isoforms.

Response: We completely agree with the reviewer's viewpoint. We have conducted the electrophysiological test on the DM1 model. For the detailed results of our electrophysiological tests, please refer to our response for Further Major Concerns #1 below.

Further major concerns:

1. The authors chose to utilize the more accessible larval neuromuscular junction (NMJ) to investigate the neuropathology of DM1. However, they did not incorporate any functional readouts, such as electrophysiology, to assess potential changes in neurotransmission associated with DM1. Instead, they focused solely on the electrophysiological phenotypes of FasII-C overexpression (OE), which cannot be considered a valid proxy for DM1. This approach significantly limits the study's ability to provide comprehensive insights into DM1 neuropathology and its impact on neurotransmission.

Response: We totally agree with the reviewer's comments. Therefore, we performed the requested electrophysiological analysis for the revision (**Fig. 2**), and it has helped us to refine our model in several aspects. We are grateful for the experimental suggestion.

Our initial analysis showed that the DM1 model had both locomotion defects and blunted synaptic development. Considering just DM1, the prediction from those results would be that neurotransmission would be diminished too. However, overexpression of *CUG₄₈₀* caused some functional phenotypes that are different from that of FasII-C overexpression. Pre+postsynaptic expression of *CUG₄₈₀* resulted in increased excitatory postsynaptic potential (EPSP) amplitude and quantal content (QC) (**Fig. 2**), whereas pre+postsynaptic expression of *UAS-FasII-C*

resulted in decreased miniature EPSP (mEPSP) amplitude, EPSP amplitude, QC and mEPSP frequency (**Fig. 6**). In our Discussion section, we hypothesized that the phenotypes observed in *CUG₄₈₀*-expressing animals could be due to the upregulation of other FasII isoforms, such as FasII-A-PEST+ and FasII-A-PEST- in addition to FasII-C.

This idea makes sense considering some prior work. Specifically, our co-authors (C. Andrew Frank) previously published a study examining overexpression of various FasII isoforms and their effects on synaptic plasticity (Spring et al., 2016). The results showed that when either of the two FasII-A isoforms was overexpressed pre- and postsynaptically, QC was increased. Furthermore, when Total-FasII was overexpressed, QC was also increased. These results indicate that the effects observed in *CUG₄₈₀*-expressing animals may represent a combinatorial effect of multiple upregulated FasII isoforms.

We note that the increased QC in the DM1 model does not necessarily indicate increased muscle activity. Rather, the increase in QC represents more vesicles being released per evoked response, which could be a compensatory mechanism to counter the functional impairment of the *CUG₄₈₀*-expressing animals' reduced locomotor activities. This idea matches prior evidence supporting the existence of robust homeostatic regulatory systems at the NMJ.

2. 2. The authors did not investigate potential signaling pathways known to mediate NMJ bouton structure, such as BMP and Wnt signaling. There are numerous established tools available in the field for studying these pathways, and the authors should utilize these to elucidate the mechanisms regulating NMJ structural phenotypes in DM1. Currently, the connection between DM1 and FasII does not sufficiently explain the observed NMJ structural phenotypes. By exploring these signaling pathways, the authors could provide a more comprehensive understanding of the molecular mechanisms underlying the structural abnormalities seen in DM1.

Response: We thank the reviewer for these insightful comments. In this revision, we have examined the impact of BMP signaling on the *Drosophila* DM1 model NMJ (**Supplementary Fig. 13**). In the *Drosophila* BMP signaling pathway, Glass Bottom Boat (Gbb) is a retrograde signaling ligand molecule that is necessary for normal synaptic growth. At the presynaptic motorneuron terminal, a receptor complex formed by Wishful Thinking (Wit) and Thickveins (Tkv) is known to be activated by Gbb to regulate this retrograde signal (Smith et al., 2012). In our experiment, we attempted to rescue the DM1 model by expressing *UAS-Gbb-GFP* or *UAS-Tkv.CA*. Tkv.CA is a constitutively active form of the Tkv receptor. Expression of neither of these constructs could rescue the decreased boutons in the DM1 background (**Supplementary Fig. 13a and 13b**). It is possible that Wnt is involved rather than Gbb. It is also possible that specific levels of Gbb signal are required for synapse integrity.

We appreciate the reviewer's suggestion on investigating the BMP and Wnt pathways. Since these are pathways that involve complex molecular mechanisms, it will take a lot more experiments to decipher which and how they participate in the DM1 model. Although we are unable to demonstrate a rescue involving the downstream retrograde signaling at the moment, we intend to further investigate these pathways in our next study. In the current study, we would like to focus on the synergistic effect of the adhesion molecule FasII in the model.

3. The authors proposed the hypothesis of “FasII isoform competition to bind to isoform A” based on the distinct phenotypes observed in NMJ boutons and larval locomotion after FasII isoform knockdown or overexpression. This hypothesis appears to be speculative and lacks robust support from the provided data and existing literature. Additionally, the potential

differing strengths of the C380 and C57 Gal4 lines and fasII RNAi lines, as shown in Figure S1b, may further complicate the situation.

Response: We apologize for the lack of support for this speculation once again and thank the reviewer for pointing this out. As the reviewer suggested, the potential differing strengths of Gal4 lines and FasII-RNAi lines may further complicate the situation. Thus, we decided to remove this interpretation from the results section but to introduce this as a possible hypothesis in the discussion section instead.

4. Using the number of NMJ boutons as the primary and nearly exclusive indicator of synaptic abnormalities is superficial, also in the context of DM1. For instance, in Figure 4, although FasII-A-PEST+ overexpression does not affect NMJ bouton numbers, it does result in a gain-of-function in locomotion. This discrepancy suggests that the NMJ bouton count alone may not fully capture the synaptic changes in DM1.

Response: We thank the reviewer's comments and agree with the reviewer's concerns. We have now added electrophysiological analysis of *CUG*₄₈₀ overexpression in motorneurons and/or muscles (Fig. 2), which further revealed the synaptic functional abnormalities of the model (Response to Further Major Concerns #1).

Apart from using NMJ boutons as the primary indicator of synaptic abnormalities, we have also used arbor disassembly to quantify synaptic abnormalities (Fig. 1d, 1e, 5c, 5d and 5h). For your convenience, we have cropped up parts of these figures below for your perusal (see below). In some cases, this phenotype appears to be more sensitive to expanded *CUG* RNA toxicity than bouton numbers, since expression of *CUG*₄₈₀ in motorneurons alone is sufficient to induce a phenotype (Fig. 1e and 5h).

Cropped from Figure 1:

Cropped from Figure 5:

Furthermore, while the authors mention the use of anti-FasII (1D4) in immunostaining methods, it is surprising that they did not perform FasII staining using isoform-specific antibodies like 1D4 versus 34B3. Conducting NMJ immunostaining against FasII would provide critical spatial information regarding FasII distribution in DM1 and FasII overexpression models. Such information would be essential for understanding the precise role of FasII isoforms in NMJ structural phenotypes and could significantly enhance the mechanistic insights derived from the study.

Response: We completely agree with the reviewer's comments and suggestions. There are two available *Drosophila* FasII antibodies that have been previously characterized: 1D4 and 34B3. 1D4 can only detect the two FasII-A isoforms (Hummel et al., 2000), while 34B3 can potentially detect all FasII isoforms (Beck et al., 2012; Grenningloh et al., 1991) (**Supplementary Fig. 8c**). We have conducted the NMJ immunostaining against FasII using 1D4 and 34B3 antibodies respectively as suggested by the reviewer (**Supplementary Fig. 9 and Fig. 10**). Unfortunately, results from 1D4 immunostaining showed no detectable changes in the spatial distribution of FasII-A in *CUG*₄₈₀-expressing animals (**Supplementary Fig. 9**). As for 34B3, no immunostaining was detected in any boutons of the NMJs at muscle 6 and 7 (**Supplementary Fig. 10a**). However, we did observe very faint signals occasionally from the terminal boutons of the NMJs at muscle 15 and 16 of segment A5 and A6 of *CUG*₄₈₀-expressing animals (**Supplementary Fig. 10b and 10c**). Such signals were not observed in the control *CUG*₆₀-expressing animals, which indicated an upregulation of FasII in the disease model. This result was consistent with our findings on *fasII* upregulation (**Fig. 4a-d**).

Minor points:

1. In the abstract the authors state, "In this study, we characterized a novel model of DM1 neuropathology by expressing untranslated expanded CUG repeats at the *Drosophila* larval neuromuscular junction." This in my eyes is misleading, as the use of CUG repeat expression to model DM1 in *Drosophila* was proposed over 15 years ago. Thus, the model itself is not that novel. Am I right here? If yes the authors should clarify this point to avoid overstating the novelty of their approach and to accurately reflect the existing body of research.

Response: We apologize for the inappropriate wording here and thank the reviewer for pointing this out. We have amended the sentence to, "Building on an established *Drosophila* model of DM1, we studied the neuropathological features of the disease by expressing untranslated expanded *CUG* repeats at the *Drosophila* larval neuromuscular junction."

2. In Figure 2, was *elav*-GeneSwitch/*C57*-driven DM1 model also sufficient to cause NMJ bouton phenotypes like with *C380/C57* ?

Response: We thank the reviewer for raising this question. We performed this experiment and demonstrated that *elav^{GS}/C57*-driven DM1 model is indeed sufficient to cause NMJ bouton phenotypes, like that of *C380/C57*. This data has now been included in **Supplementary Fig. 2**.

3. Quantification is missing in Figure 3a,b.

Response: We apologize for this missing part and thank the reviewer for pointing this out. The original **Fig. 3** has become **Fig. 4** in the revised manuscript. We have now included the quantifications (**Fig. 4b and 4d**) for this figure.

4. In Figure 3c,d, the NMJ bouton phenotypes and larval locomotion of *fasII-A* and *A&C RNAi* are shown only in DM1 background, the effects of *fasII-A* and *A&C RNAi* alone are missing

Response: We thank the reviewer for pointing this out. We apologize for the missing part and have included the NMJ bouton phenotypes and larval locomotion analysis of overexpressing *fasII-A-RNAi* and *fasII-A&C-RNAi* in the wild type background now (**Supplementary Fig. 5**).

5. In Figure 4b, *CUG-60* expression is not the correct control for *FasII OE*

6. Similar in Figure 4d,e, *CUG-60* expression is not the correct control for *FasII-C OE*

Response: We apologize for not having the correct controls and thank the reviewer very much for pointing this out. We have now added the controls (depicted as white bars on the graphs) into the revised **Fig. 5b, 5d and 5e** (the original **Fig. 4** has become **Fig. 5** in the revised manuscript).

7. In Figure 5, the authors differentiated pre- and post- OE of *FasII-C* in electrophysiology. However, the authors did do the same in Figure 4.

Response: We thank the reviewer for his/her comments. We have now performed NMJ bouton phenotypical and larval locomotor analysis on pre- and post-overexpression of *FasII-C* animals. The data is now shown in the revised **Fig. 5f, 5g and 5h**. Our results showed that overexpression of *FasII-C* using a single driver (either *C380* or *C57* alone) was not sufficient to cause significant bouton changes at the NMJ (**Fig. 5g**). However, *C380* alone was sufficient to cause arbor disassembly (**Fig. 5h**), and either *C380* or *C57* alone was sufficient to cause locomotor defects (**Fig. 5i**). These results very closely resemble overexpression of *CUG₄₈₀* (**Fig. 1**). Furthermore, they also showed a synergistic effect between pre- and postsynaptic expression.

8. In Figure 6, did the authors also performed fasII-C OE in DM1? Were there additive effects in NMJ bouton number and locomotion, and electrophysiology

Response: We thank the reviewer for raising this concern. We have now performed NMJ bouton phenotypical and larval locomotor analysis to investigate the effect of FasII-C overexpression in DM1 background. Our results showed that the combined effect of FasII-C and *CUG₄₈₀* overexpression resulted in significant decreases of number of boutons and locomotor activity in comparison to *CUG₄₈₀* overexpression alone (**Fig. 7a-c**) (the original **Fig. 6** has become **Fig. 7** in our revised manuscript). This suggests that the upregulation of FasII-C in the DM1 model may not be at saturation point yet, and that further increase of FasII-C results in more severe phenotypes.

After performing electrophysiological analysis of *CUG₄₈₀* overexpression (**Fig. 2**), we showed that the transmission phenotypes of these animals are in fact opposite to FasII-C overexpression. Thus, it would not be possible to have an observable additive effect using this method of analysis.

9. Except for DM1 model flies, genetic controls for all the UAS lines including RNAi lines and cDNA OE line are missing. Given that genetic background affects NMJ bouton phenotypes, it will be important to include those UAS controls.

Response: We agree with the reviewer and thank him/her for this comment. We have now included the bouton phenotype analysis of these genetic controls in **Supplementary Fig. 21**. No significant differences were observed among the analyzed control genotypes.

10. It is also a bit surprising that, given the per se limited electrophysiological data of Figure 5, the authors did not show any sample traces for minis and evoked.

Response: We thank the reviewer for pointing this out and apologize for the missing part. We have now added the sample traces into the revised **Fig. 6 (Fig. 6m, 6n and 6o)** (the original **Fig. 5** has become **Fig. 6** in the revised manuscript). Representative traces have also been included in the new **Fig. 2** (electrophysiological result of *CUG₄₈₀*-expressing DM1 model). In summary, overexpression of FasII-C results in decreased transmission, while overexpression of *CUG₄₈₀* (which causes upregulation of FasII-A-PEST+, FasII-A-PEST- and FasII-C) results in increased transmission.

Reviewer #2 (Remarks to the Author):

Inconsistencies :

1. Authors refer in the title and in the text to neuropathological and behavioral defects in DM1 but they analyse phenotypes of neuromuscular junctions and larval mobility. Whether dysregulation of FasII influences fly behavior and neurological CNS functions is not shown.

Response: We thank the reviewer for raising concern about possible inconsistencies. In response to this concern, we have amended our paper's title to "Pre- and postsynaptic upregulation of FasII synergistically underlies neuropathological and behavioral phenotypes in a *Drosophila* model of myotonic dystrophy". In the text of the manuscript, we have also substituted the word "defects" with "phenotypes".

Regarding the concern of the dysregulation of FasII influencing fly behavior and CNS functions, we have shown that the knockdown of FasII in the DM1 model fully rescued both NMJ and behavioral phenotypes (**Fig. 4e-i**). Also, the overexpression of FasII-C causes NMJ phenotypes and behavior impairment that resemble the DM1 model (**Fig. 5a-i**). Furthermore, the overexpression of FasII-C directly influences synaptic transmission from motoneurons (motoneurons are part of the CNS, so motoneuron functions are CNS functions) (**Fig. 6a-o**). Last but not least, the NMJ and behavior phenotypes of the DM1 model could be fully rescued by overexpressing FasII-A-PEST+ (**Fig. 7a-g**). All these experiments strongly suggest that the dysregulation of FasII influences *Drosophila* behavior and CNS functions.

2. On page 4 authors say that they overexpressed control UAS-CUG60 construct (which should rather be UAS-CTG60) with the number of CTG repeats that is lower than the disease threshold. This is not correct as the number of CTG repeats in *dmpk* gene higher than 37 is already classified as potentially DM1 pathogenic.

Response: We thank the reviewer for his/her comments. While it is true that the constructs we used were *UAS-CTG₆₀* and *UAS-CTG₄₈₀*, we would like to emphasize to the readers that *CUG₆₀* and *CUG₄₈₀* were the untranslated transcriptional products (RNA) in our experiments that drove our results. Thus, we used “*CUG₆₀*” or “*CUG₄₈₀*” in most parts of the manuscript. We are glad that the reviewer reminded us to clarify this difference in naming more clearly.

As for the disease threshold of DM1, even though 37 is already classified as “potential for DM1”, we were unable to detect any phenotype in our experiments. As shown in **Fig. 5b and 5e**, *CUG₆₀*-expressing animals showed no phenotypes in comparison to driver controls regarding both NMJ morphology and locomotor behavior. The *UAS-CTG₆₀* line was previously characterized and published (Garcia-Lopez et al., 2008), and we have not found any abnormalities in *CUG₆₀*-expression animals in our study.

3. On page 6 authors comment that « Previous studies suggested that the expression of CUG480 in the postsynaptic BWMs alone was sufficient to produce locomotor defects in *Drosophila* larvae ». They refer to Picchio et al., 2013.

This is not correct. In this paper larval locomotion was analysed using inducible UAS-240, UAS-600 and UAS-960 CTG repeat lines but not UAS-480. Also in Picchio et al., repeats expression was driven by Mef-GAL4 showing much earlier muscle expression than C57-GAL4 used by authors.

Response: We apologize for this error and thank the reviewer so much for pointing this out. We have now amended the sentence to, "A previous study demonstrated that the expression of expanded *CUG* repeats in the postsynaptic BWMs alone was sufficient to produce locomotor defects in *Drosophila* larvae (Picchio et al., 2013)."

In addition, we cited this landmark paper again in the Introduction section to highlight its significant contribution to the development of DM1 models in *Drosophila*.

General comment :

Authors hypothesize that FasII could be dysregulated by CUG repeat transcripts, Underlying question is whether this dysregulation is dependent on Mbn1, major RNA-binding protein sequestered by toxic CUG repeats.

Muscle targeted knockdown of Muscleblind (Mbl), *Drosophila* ortholog of Mbn1, mimics partial loss of Mbn1 and could be used to test whether NMJ phenotypes and FasII deregulation is Mbl-dependent.

Response: We thank the reviewer very much for raising this question. We agree with the underlying question regarding the involvement of Mbl in the dysregulation of FasII. Therefore, we conducted the analysis on Mbl in our fly model based on the advice of the reviewer.

In *Drosophila*, we analyzed *mbl*^{E27} mutants to determine whether they recapitulated the *CUG*₄₈₀ phenotypes. *mbl*^{E27} is a homozygous lethal null mutation (Garcia-Lopez et al., 2008). Therefore, heterozygous *mbl*^{E27} mutants were analyzed in the investigation. The heterozygous *mbl*^{E27} mutants have a bouton reduction phenotype and an arbor disassembly phenotype that indeed recapitulated *CUG*₄₈₀ (**Supplementary Fig. 14a-c**).

In addition to NMJ analysis, we performed RT-PCR on *mbl* mutants to examine their expression of *fasII*. Our results showed that the heterozygous *mbl*^{E27} mutants exhibited upregulated *fasII*, similar to *CUG*₄₈₀ overexpression (**Supplementary Fig. 14d and 14e**). These data suggested that the partial loss of Mbl functions in the heterozygous *mbl*^{E27} mutant might reflect the physiological conditions of the DM1 model. Furthermore, these data also supported Mbl's role in *fasII* processing and NMJ morphology in the DM1 neuropathological context.

Specific comments:

Fig. 1

- Fig. 1a - In the presented view the reduction of bouton number in CUG480 versus CUG60 is not evident to see....A larger view showing NMJs of muscle 6 and 7 would help to appreciate the reduction in bouton numbers and the retraction of arbors.

Response: We thank the reviewer for the comment. We would like to explain that an image of the entire NMJ (lower magnification) would help the reader to appreciate the reduction in bouton numbers, whereas a magnified (larger) view would help the reader to appreciate the retraction of arbors. **Fig. 1a** aimed to present the entire NMJ of muscle 6 and 7, so that the reader can more easily observe the reduction of total bouton numbers. On the other hand, a magnified (larger) view of the boutons is presented in **Fig. 1d** to allow the reader to more clearly see the retraction of arbors.

- It is surprising that an increase in arbor disassembly observed in C380-driven presynaptic overexpression of CUG480 does not reduce the bouton number.

Response: We thank the reviewer for his/her comment. This may be due to the methods of quantification. When quantifying the number of NMJs with disassembling arbors, an NMJ with even just one disassembling arbor is regarded as an NMJ with disassembly. However, since each NMJ has many arbors, it is possible that in some cases, one disassembling arbor is not sufficient to impact on overall bouton numbers.

- Criteria for quantification of arbors disassembly are not clear. Segmentation and image analyses tools allowing to measure arbor's surface/extensions need to be applied to assess NMJ arborisations in different genetic contexts.

Response: We apologize for overlooking this and thank the reviewer for pointing this out. We have now added the details of our quantification method in the Methods section. In brief, a disassembling arbor was defined as a stretch of at least two boutons that are severed from the arbor and from one another (immunostained by anti-HRP), in which at least one of these boutons showed DLG staining on the postsynaptic muscle. would be regarded as a "disassembling arbor", while any NMJ with at least one disassembling arbor would be counted as an "NMJ with disassembling arbors".

We previously explored the possibility of conducting our quantification using 3D image analyses tools, such as IMARIS or AIVIA. However, these tools are unable to definitively recognize disassembling arbors according to our criteria. Thus, we performed our quantification manually.

- Fig. 1d – Dlg signal is saturated. A larger view of NMJs is required.

Response: We thank the reviewer for his/her comment. It is known that the Dlg signal of type I small boutons (type Is) is much fainter than that of type I big boutons (type Ib) (Koh et al., 2000). In **Fig. 1d**, we had to saturate the signals on type Ib boutons to visualize the dim Dlg signals of type Is of both genotypes. In particular, the remaining Dlg signals from type Is in *C380+C57>CUG₄₈₀* animals are what we would like to show the readers. Therefore, we cannot reduce this saturation on the image. The confocal images shown in **Fig. 1d** are already magnified images of the NMJs. Whole NMJ view of the same genotypes are shown in **Fig. 1a**.

- It would be appropriate to present expression pattern of the two drivers (C57 and C380) used in this study (in a supplementary data).

Response: We thank the reviewer's comment. We have now included confocal micrographs to show the expression pattern of C380 and C57 (**Supplementary Fig. 20**). Further details regarding the expression pattern of *C380-Gal4* can be found in Sanyal in 2009 (Sanyal, 2009). In short, in *C380-Gal4*, Gal4 is expressed in most larval motor neurons, dorsal medial cluster and larval brain lobes. Similar to *C380*, the expression pattern of *C57* has also been presented in several previous studies (Budnik et al., 1996; Gorczyca et al., 2007; Zhao & Geisbrecht, 2025). In brief, in *C57-Gal4*, Gal4 is expressed in all larval muscles, from mid-first to third instar stage.

Fig. 2

Considering that several FasII isoforms are produced and differentially dysregulated in DM1 context, western blot would allow to visualize isoform specific dysregulation. Using slot blot and not western blot when assessing FasII protein expression in *Drosophila* is surprising.

Response: We thank the reviewer for raising this concern. There are only two available FasII antibodies: 1D4 and 34B3. 1D4 only detects the two FasII-A isoforms (Hummel et al., 2000), while 34B3 is predicted to detect all FasII isoforms (Beck et al., 2012; Grenningloh et al., 1991) (**Supplementary Fig. 8c**). Since 1D4 only detects FasII-A, only 34B3 has the potential to detect and differentiate different FasII isoforms in WB. The possible band sizes for the different FasII isoforms are: FasII-A-PEST+ (96 kDa), FasII-A-PEST- (93 kDa), FasII-C (90 kDa), FasII-B (86 kDa). However, we found that 34B3 could only allow us to effectively visualize a single band in *Drosophila* CNS and muscles even when we over-exposed the blots (**Supplementary Fig. 11**). This band likely represented FasII-A-PEST+, which is the major isoform known to be expressed in the CNS (Wright & Copenhaver, 2001). Even though 34B3 might have recognized other isoforms, there were just insufficient signals for other isoforms to be visualized via WB. Therefore, we resorted to the slot blot approach to detect all isoforms together (**Fig. 3e and 3f**), while relying on RT-PCR instead to analyze individual *fasII* isoforms at the transcript level.

FasII ortholog NCAM1 in DM1 mouse models and in DM1 patients is also upregulated. Is NCAM1 presenting several isoforms that are differentially dysregulated in DM1?

Response: We thank the reviewer for raising this question. To determine whether NCAM1 isoforms are dysregulated in DM1, we performed semi-quantitative RT-PCR analysis of alternative splicing using cultured astrocytes and frontal cortex tissues of DMSXL mice. Our

results showed that there was indeed splicing dysregulation of *Ncam1* transcripts in cultured astrocytes and in the frontal cortex of DMSXL mice (**Supplementary Fig. 17**).

Fig. 3

This figure would benefit from a scheme of FasII gene with different FasII isoforms. According to Flybase, FasII encodes 7 different isoforms. Why authors focus on FasII A and FasII C only? If alternative splicing is in play, how FasII splice variants are expressed in DM1 contexts.

Response: We thank the reviewer for raising this concern. Although there are 7 different isoforms in Flybase (Fas2-RA, Fas2-RB, Fas2-RC, Fas2-RD, Fas2-RF, Fas2-RG and Fas2-RH), most are only predicted, and have not been shown to be expressed in neuronal or muscular tissues of the larva. We did not dismiss the possibility that there could be other isoforms expressed at the NMJ. But instead of investigating all potential isoforms, it would be more reasonable to initially focus on FasII-A-PEST+, FasII-A-PEST- and FasII-C, as they have been characterized and tools are available (Wright & Copenhaver, 2001). We have now included a schematic diagram of these FasII isoforms (**Supplementary Fig. 8**). Since we were able to rescue the DM1 phenotypes by either knocking down Total-FasII or FasII-A&C, but not by knocking down FasII-A, it was more logical to further pursue studying FasII-C rather than investigating other uncharacterized isoforms.

As a side note, out of the 7 predicted isoforms in Flybase (RA, RB, RC, RD, RF, RG and RH), Fas2-RA, Fas2-RB and Fas2-RC are NOT alternatively names for FasII-A, FasII-B and FasII-C. The actual translations are as follows:

FasII-A-PEST+ = Fas2-RA

FasII-A-PEST- = Fas2-RC

FasII-C = Fas2-RB

We have now clarified this in **Supplementary Fig. 8**.

In contrast to FasII-A-PEST+ and FasII-A-PEST- which are known to be expressed at the *Drosophila* nervous system and neuromuscular junction to mediate intercellular interactions (Wright & Copenhaver, 2001), FasII-C was proposed to act as homotypic bridging proteins within the same renal Malpighian tubule cell to stabilize the microvillar brush border of against shear stress (Halberg et al., 2016).

Fig.4 Simultaneous overexpression of FasII C in MNs and BWMs leads to arbors disassembly. Did authors check effects of FasII C overexpression in MNs only?

Response: We thank the reviewer for his/her insightful comment. We have now performed the arbor disassembly analysis on animals expressing FasII-C in either motorneurons or body wall muscles and included it in our revised **Fig. 5**. Like expressing *CUG₄₈₀*, FasII-C overexpression in motorneurons only (*C380*) resulted in a small but significant increase of arbor disassembly, while FasII-C overexpression in body wall muscles only (*C57*) did not have any impact (**Fig. 5h**) (the original **Fig. 4** has become **Fig. 5** in the revised manuscript). For your convenience, we have cropped the corresponding part of the figure below for your perusal (**see below**).

Cropped from Figure 5:

Fig.6 – Link between boutons number and locomotor capacities is unclear. Why increase in boutons number in FasII A-PEST- rescue context does not result in mobility improvement ?

Response: We thank the reviewer for raising this question. Usually, the number of boutons correlate well with locomotor activity of an animal. However, it also depends on whether the morphology of the boutons is normal. We analyzed the boutons' morphology of the FasII-A-PEST- rescue animals and found abnormalities (**Fig. 7h and 7i**) (the original **Fig. 6** has become **Fig. 7** in the revised manuscript). Normal boutons are supposed to have the well-known "beads-on-a-string" morphology, which can be seen in the FasII-A-PEST+ rescue animals. However, many boutons of the FasII-A-PEST- rescue animals do not have such round shape. We speculate that these abnormalities might have contributed to abnormal functions in the FasII-A-PEST- animals.

Reviewer #3 (Remarks to the Author):

This is a very nice paper with interesting phenotypes and genetics. In this study, the authors reveal the function of fasII in disassembly and impairment of locomotor activity in the *Drosophila* DM1 model. While the manuscript is well-written and likely to contribute to the field, it could be further improved with additional data and analyses, which are crucial for strengthening the conclusions.

Response: We would like to thank the reviewer for the compliment. We appreciate the time and effort of the reviewer to help us improve the manuscript.

1a) The connection to DM1 is tenuous. NCAM1 changes in DM1 seem variable depending on what tissue or dataset they are looking at.

Response: We thank the reviewer for raising this concern. While NCAM1 could be downregulated in certain tissues of DM1 patients (**Supplementary Fig. 12**), it was upregulated in most of the mammalian tissues that we have examined (**Fig. 3i-n**). Furthermore, regardless of whether NCAM1 is upregulated or downregulated, as long as NCAM1 is dysregulated, it is

likely to impair synapse functions and integrity, which may reflect the synaptopathology in DM1.

1b) Also, there is no clarity to what these changes mean.

Response: We thank the reviewer for this comment. Since we found that NCAM1 is consistently dysregulated in DMSXL mice and DM1 patients (across different types of CNS and muscular tissues), and this dysregulation could impair neuronal and muscular functions, it is likely that the NCAM1 changes we observed will provide insights into the neural pathogenesis of DM1.

Our findings also bridge the well-known “upstream” pathology of ribonuclear foci sequestration of splicing factors over to the largely unexplored “downstream” pathology of synapse dysfunction in the disease.

In our study, we have proposed a working model of synaptopathology in the NMJ model of DM1 (**Fig. 8**), which explained how the changes in FasII/NCAM1 contribute to the loss of synapses.

1c) Similar things can be said about the data from DMSXL mice. The data doesn't say anything about where the NCAM is being expressed (what cells, etc) and so has no relevance in the context of this manuscript.

Response: We thank the reviewer for raising this question. Regarding cell types that express NCAM1, in our additional experiments performed in this revision, we were able to demonstrate *Ncam1* splicing dysregulation in cultured mouse astrocytes of DMSXL mice (**Supplementary Fig. 17a-c and 18a-b**), implying that NCAM1 is at least expressed in this cell type. In general, NCAM1 is known to be expressed in most neuronal cell types, thus we mostly used mouse tissues such as the spinal cord or the frontal cortex to examine NCAM1 changes (**Fig. 3i, 3k, 3m, Supplementary Fig. 17 and 18**). These results demonstrated the relevance of NCAM1 dysregulation in the context of neuropathology of DM1.

2) There is no data on the expression level of the CUG transgene in the various conditions. This is essential to ensure that this element of the experiment is controlled and assessed.

Response: We thank the reviewer for raising this concern. We conducted RT-PCR to investigate the *CUG* transgene expression level in the CNS and muscle tissues of *Drosophila* 3rd instar larvae respectively (**Supplementary Fig. 19**). Since forward primers containing *CTG* repeats and reverse primers containing *CAG* repeats would freely bind to anywhere along the region of *CTG* repeats, resulting in variable band sizes, we used primers against the *heat shock promoter (hs)* and primers against the *simian vacuolating virus 40 (SV40)* instead. These sequences are present in the expressed transcripts of *UAS-CTG₆₀* and *UAS-CTG₄₈₀* constructs. The *hs* primers were used to detect the *CUG* transgene expression in the CNS tissues because *hs* was absent in *elav^{GS}-GAL4*. Meanwhile the *SV40* primers were used to detect the *CUG* transgene expression in the muscle tissues, because *SV40* was absent in *C57-GAL4*. The forward and reverse primer sequences of the *hs* and *SV40* were described in the Methods section. Both *CUG₆₀* and *CUG₄₈₀* were well-expressed in the CNS and muscle respectively (**Supplementary Fig. 19**). Further characterizations of the *UAS-CTG₆₀* and *UAS-CTG₄₈₀* fly lines could be found in Garcia-Lopez et al., 2008 and Luu et al., 2016 (Garcia-Lopez et al., 2008; Luu et al., 2016).

3) Do the CUG repeats get expressed in the NMJ and which cells if any have RNA foci? For example, in Figure 1, although it has been shown previously that CUG480 RNA is expressed in this model, fluorescence in situ hybridization (FISH) analysis will help show that CUG480 RNA

is also expressed in the pre-synaptic MNs and post-synaptic BWMs. Does Mbl get sequestered in those cells? Is Mbl expressed in the pre-synaptic and post-synaptic cells? Does the RNA get expressed there?

Response: We thank the reviewer for his/her insightful comment. We have now performed Fluorescence In Situ Hybridization (FISH) analysis on *Drosophila* larvae according to the reviewer's suggestion. The result has shown that *CUG₄₈₀* RNA is present as foci in the nuclei of both pre-synaptic motorneurons and post-synaptic body wall muscles (**Supplementary Fig. 1**). A previous study demonstrated that exogenous MblC:GFP forms nuclear inclusions with the ribonuclear foci in the muscle nuclei of the *Drosophila* DM1 model (Garcia-Lopez et al., 2008). Another study also demonstrated that endogenous Mbl co-localizes with *CUG*-containing ribonuclear foci (de Haro et al., 2006). Unfortunately, due to the limitation of the antibodies, we were unable to perform immunostaining of Mbl in our samples. However, according to Li et al., 2022, Mbl is highly expressed in both motorneurons and body wall muscles (Li et al., 2022).

4) Does an Mbl knockout recapitulate the CUG480 phenotype? Does it or does it not? Answer would be interesting either way.

Response: We thank the reviewer for raising this insightful question. We agree with the answer would be interesting in either way. Hence, we conducted the analysis on Mbl using our fly model.

In *Drosophila*, we analyzed *mbl^{E27}* mutant to determine whether they recapitulated the *CUG₄₈₀* phenotypes. *mbl^{E27}* is a homozygous lethal null mutation (Garcia-Lopez et al., 2008). Therefore, heterozygous *mbl^{E27}* mutants were analyzed in the investigation. The heterozygous *mbl^{E27}* mutants have a bouton reduction phenotype and an arbor disassembly phenotype that indeed recapitulated *CUG₄₈₀* (**Supplementary Fig. 14a-c**).

In addition to NMJ analysis, we performed RT-PCR on *mbl* mutants to examine their expression of *fasII*. Our results showed that the heterozygous *mbl^{E27}* mutants exhibited upregulated *fasII*, similar to *CUG₄₈₀* overexpression (**Supplementary Fig. 14d and 14e**). These data suggested that the partial loss of Mbl functions in the heterozygous *mbl^{E27}* mutant might reflect the physiological conditions of the DM1 model. Furthermore, these data also supported Mbl's role in *fasII* processing and NMJ morphology in the DM1 neuropathological context.

5) There is no description of how the quantification of FasII RNA levels were carried out. Was it slot blot or quantitative real-time RT-PCR. The first method is inadequate. This study's findings depend on the upregulation of *fasII* (cell adhesion molecule) in different models. If done quantitatively, the upregulation of *fasII* data could be more convincing (Figure 2). Authors need to verify the upregulation of *fasII* by different methodologies, too.

Response: We apologize for this missing part and thank the reviewer for pointing this out. The original **Fig. 3** has become **Fig. 4** in the revised manuscript. We have now included the quantifications (**Fig. 4b and 4d**) for this figure, and the quantification method has now been included in the Methods section.

For RT-PCR, we chose not to perform quantitative PCR, but to perform semi-quantitative RT-PCR instead. The reason for this is that *fasII-A-PEST⁻* is only missing a very small PEST domain compared to *fasII-A-PEST⁺* (**Supplementary Fig. 8a**), so the primers for *fasII-A-PEST⁻* will always pick up *fasII-A-PEST⁺* as well (Beck et al., 2012). Since *fasII-A-PEST⁺* and *fasII-A-PEST⁻* are always amplified in the same reaction, it is difficult to investigate using real-time

RT-PCR primers as the increase of fluorescence represents the combined increase of both isoforms.

In summary, we verified the upregulation of FasII in *Drosophila* with 2 methods: (1) by semi-quantitative RT-PCR (mRNA) (**Fig. 3a-d and Fig. 4a-d**), and (2) by slot-blot (protein) (**Fig. 3e-h**).

6) There is an implication in this study that there is an imbalance in the isoforms of FasII. No clear studies were done to assess the relative levels of the isoforms at both the levels of the RNA and protein. This would be essential.

Response: We thank the reviewer for raising this concern. To assess the relative levels of the *fasII* isoforms in the context of RNA, we have performed semi-quantitative RT-PCR in the *Drosophila* CNS and muscle tissues respectively. In **Fig. 4a and 4b**, the *Total fasII* and three targeted *fasII* isoforms, *fasII-A-PEST+*, *fasII-A-PEST-* and *fasII-C*, are upregulated in the *CUG₄₈₀*-expressing *Drosophila* CNS. Consistent with the CNS, the *Total fasII* and all the three targeted *fasII* isoforms are upregulated in the *Drosophila* muscle as showed in **Fig. 4c and 4d**. These results have shown that the *fasII* isoforms are dysregulated in the *CUG₄₈₀*-expressing fly model at RNA level.

We tried to assess the FasII isoforms protein levels using Western blot (WB). There are only two available FasII antibodies: 1D4 and 34B3. 1D4 only detects the two FasII-A isoforms (Hummel et al., 2000), while 34B3 is predicted to detect all FasII isoforms (Beck et al., 2012; Grenningloh et al., 1991) (**Supplementary Fig. 8c**). Since 1D4 only detects FasII-A, only 34B3 has the potential to detect and differentiate different FasII isoforms in WB. The possible band sizes for the different FasII isoforms are: FasII-A-PEST+ (96 kDa), FasII-A-PEST- (93 kDa), FasII-C (90 kDa), FasII-B (86 kDa). However, we found that 34B3 could only allow us to effectively visualize a single band in *Drosophila* CNS and muscles even when we over-exposed the blots (**Supplementary Fig. 11**). This band likely represented FasII-A-PEST+, which is the major isoform known to be expressed in the CNS (Wright & Copenhaver, 2001). Even though 34B3 might have recognized other isoforms, there were just insufficient signals for other isoforms to be visualized via WB. Therefore, we resorted to the slot blot approach to detect all isoforms together (**Fig. 3e and 3f**), while relying on RT-PCR instead to analyze individual *fasII* isoforms at the transcript level in *Drosophila* (**Fig. 4a-d**).

7) FasII protein expression at the NMJ/boutons should have been done using isoform specific and total FasII antibodies, as described in reference 39. For example, Figure 3 shows the rescues of NMJs in the *Drosophila* DM1 model, but no data shows how much knockdown was achieved.

Response: We thank the reviewer for his/her insightful comment. Since we were unable to use 34B3 to pick up different FasII isoforms in our WB, we opted for the alternative method of RT-PCR to demonstrate the level of knockdown achieved with different RNAi constructs. The levels of FasII knockdown for each isoform by FasII-A&C-RNAi was shown in **Supplementary Fig. 3** (the line FasII-C-RNAi#38 was eventually renamed as FasII-A&C-RNAi). For Total-FasII-RNAi and FasII-A-RNAi, the lines were characterized in reference 39 (Beck et al., 2012).

On the other hand, we performed additional NMJ immunostaining of FasII using 1D4 and 34B3 antibodies respectively (**Supplementary Fig. 9 and 10**). The results from 1D4 (detecting FasII-A only) immunostaining showed no detectable changes of FasII-A in the NMJ of *CUG₄₈₀*-expressing animals (**Supplementary Fig. 9**). For 34B3, no immunostaining was detected in any boutons of the NMJs at muscle 6 and 7 (**Supplementary Fig. 10a**). However, we did observe

very faint signals occasionally from the terminal boutons of the NMJs at muscle 15 and 16 in segment A5 and A6 of *CUG₄₈₀*-expressing animals (**Supplementary Fig. 10b and 10c**). Such signals were not observed in the control *CUG₆₀*-expressing animals, which indicated an upregulation of FasII in the disease model. This result was consistent with our findings on FasII upregulation using RT-PCR and slot blot (**Fig. 3a-h and Fig. 4a-d**).

8) Similarly, western blots should have been done to assess the relative protein levels of the various isoforms.

Response: We thank the reviewer for his/her comment. As mentioned previously (in Point #6), it was unfortunate that we were unable to assess the relative protein levels of the various FasII isoforms. Therefore, we resorted to the slot blot approach to detect all isoforms together, while relying on RT-PCR instead to analyze individual *fasII* isoforms at the transcript level (**Fig. 4a-d**).

9) The studies done by Beck et al (ref 39), identifying *beag* and *dsmu1* mutants and the essential role of these proteins in splicing of FasII, are highly relevant to this current study. They aren't even mentioned, other than a passing reference to the fact that the flies used in that paper were used in this study. The phenotypes mentioned in that paper are highly reminiscent of the results from the current study. This includes the importance of the FasII-A-PEST+ isoform for regulation of NMJ growth and bouton number. This can't be ignored. The authors are trying to make a connection to splicing defects in DM1, but ignoring the splicing factors that have already been reported to be relevant in DM1 and in FasII splicing.

Response: We thank the reviewer very much for raising this important matter. Indeed, Beck et al. has shown that *Beag* and *Dsmu1* are involved in the splicing of FasII-A, and they showed the importance of FasII-A-PEST+ in regulating NMJ bouton numbers. We have now included this in our Discussion section to recognize their findings and compare them to our results.

10a) Are there antibodies for *Beag*? If so, what are the protein levels of *Mbl*, *Elav* (the fly homolog of ELAV proteins in mammals), *Beag* and *dsmu1* in the current experiments?

Response: We are grateful to the reviewer for his/her insightful comments. We contacted our coauthors of Beck et al. (Ref 39), but unfortunately their *Beag* antibody is no longer available. Similarly, no antibodies are available for *Dsmu1*. As for *Mbl*, although we were able to acquire antibodies from our coauthor, the Artero Lab, in Spain, we were unable to produce satisfactory immunostaining.

Although we were unable to assess the protein levels, we assessed the transcriptional level by performing semi-quantitative RT-PCR as an alternative to examine *elav*, *mbl*, *beag* and *dsmu1* in the *CUG₄₈₀* fly model. Our results showed that the expression of *beag* and *dsmu1* were upregulated in *CUG₄₈₀*-expressing animals, while the expression of *elav* and *mbl* had no significant changes (**Supplementary Fig. 15**).

10b) Does *Beag* co-localize with the RNA foci and *Mbl*? Does FLAG-tagged *Beag* (see Ref 39) co-localize with the RNA foci or *Mbl*? What are the expression levels of *Beag* and *dsmu1* in the *CUG₄₈₀* model, at the RNA and protein level? The mammalian homolog of *Beag* is called RED (if I am correct) and there are antibodies for that.

Response: We thank the reviewer for the suggestions on investigating colocalization of these splicing factors with RNA foci. Although we were unable to acquire *Beag* and *Dsmu1* antibodies to conduct this experiment in flies, nor could we acquire *UAS-Beag-FLAG* or *UAS-Dsmu1-FLAG*

fly lines from our coauthors of Beck et al., we were able to acquire RED and SMU1 antibodies as the reviewer suggested. RED and SMU1 are the mammalian homolog of Beag and Dsmu1 in *Drosophila*. Thus, as an alternative, we performed immuno-FISH on primary astrocytes of DMSXL mice to examine if RED and SMU1 co-localized with CUG RNA foci. Our results showed that RED and SMU1 do not co-localize with RNA foci (**Supplementary Fig. 16**).

Summarizing the above results, although Beag and Dsmu1 are upregulated in the *Drosophila* DM1 model, RED and SMU1 do not co-localize with RNA foci in the mouse DM1 model. It is possible that the expanded *CUG* RNA caused an abnormal upregulation of these genes, which contributes to part of the pathogenesis. It is also possible that the upregulation of Beag and Dsmu1 may serve as a compensatory mechanism to alleviate the damaging effect of Mbl and FasII dysregulation. Since RED and SMU1 do not co-localize with RNA foci in the nuclei, the latter explanation seems more likely.

There are a couple of grammatical errors in the abstract section and elsewhere.

Response: We thank the reviewer for pointing this out and apologies for the grammatical errors. They have been fixed in this revised version.

Reviewer #4 (Remarks to the Author):

I co-reviewed this manuscript with one of the reviewers who provided the listed reports as part of the Nature Communications initiative to facilitate training in peer review and appropriate recognition for co-reviewers.

Response: We appreciate the effort of all reviewers in facilitating the review process of our paper.

References:

- Beck, E. S., Gasque, G., Imlach, W. L., Jiao, W., Jiwon Choi, B., Wu, P. S., Kraushar, M. L., & McCabe, B. D. (2012). Regulation of Fasciclin II and synaptic terminal development by the splicing factor beag. *J Neurosci*, 32(20), 7058-7073. <https://doi.org/10.1523/JNEUROSCI.3717-11.2012>
- Budnik, V., Koh, Y. H., Guan, B., Hartmann, B., Hough, C., Woods, D., & Gorczyca, M. (1996). Regulation of synapse structure and function by the Drosophila tumor suppressor gene dlG. *Neuron*, 17(4), 627-640. [https://doi.org/10.1016/s0896-6273\(00\)80196-8](https://doi.org/10.1016/s0896-6273(00)80196-8)
- de Haro, M., Al-Ramahi, I., De Gouyon, B., Ukani, L., Rosa, A., Faustino, N. A., Ashizawa, T., Cooper, T. A., & Botas, J. (2006). MBNL1 and CUGBP1 modify expanded CUG-induced toxicity in a Drosophila model of myotonic dystrophy type 1. *Hum Mol Genet*, 15(13), 2138-2145. <https://doi.org/10.1093/hmg/ddl137>
- Garcia-Lopez, A., Monferrer, L., Garcia-Alcover, I., Vicente-Crespo, M., Alvarez-Abril, M. C., & Artero, R. D. (2008). Genetic and chemical modifiers of a CUG toxicity model in Drosophila. *PLoS One*, 3(2), e1595. <https://doi.org/10.1371/journal.pone.0001595>
- Gorczyca, D., Ashley, J., Speese, S., Gherbesi, N., Thomas, U., Gundelfinger, E., Gramates, L. S., & Budnik, V. (2007). Postsynaptic membrane addition depends on the Discs-Large-interacting t-SNARE Gtaxin. *J Neurosci*, 27(5), 1033-1044. <https://doi.org/10.1523/JNEUROSCI.3160-06.2007>
- Grenningloh, G., Rehm, E. J., & Goodman, C. S. (1991). Genetic analysis of growth cone guidance in Drosophila: fasciclin II functions as a neuronal recognition molecule. *Cell*, 67(1), 45-57. [https://doi.org/10.1016/0092-8674\(91\)90571-f](https://doi.org/10.1016/0092-8674(91)90571-f)
- Halberg, K. A., Rainey, S. M., Veland, I. R., Neuert, H., Dornan, A. J., Klambt, C., Davies, S. A., & Dow, J. A. (2016). The cell adhesion molecule Fasciclin2 regulates brush border length and organization in Drosophila renal tubules. *Nat Commun*, 7, 11266. <https://doi.org/10.1038/ncomms11266>
- Hummel, T., Krukkert, K., Roos, J., Davis, G., & Klambt, C. (2000). Drosophila Futsch/22C10 is a MAP1B-like protein required for dendritic and axonal development. *Neuron*, 26(2), 357-370. [https://doi.org/10.1016/s0896-6273\(00\)81169-1](https://doi.org/10.1016/s0896-6273(00)81169-1)
- Koh, Y. H., Gramates, L. S., & Budnik, V. (2000). Drosophila larval neuromuscular junction: molecular components and mechanisms underlying synaptic plasticity. *Microsc Res Tech*, 49(1), 14-25. [https://doi.org/10.1002/\(SICI\)1097-0029\(20000401\)49:1<14::AID-JEMT3>3.0.CO;2-G](https://doi.org/10.1002/(SICI)1097-0029(20000401)49:1<14::AID-JEMT3>3.0.CO;2-G)
- Li, H., Janssens, J., De Waegeneer, M., Kolluru, S. S., Davie, K., Gardeux, V., Saelens, W., David, F. P. A., Brbic, M., Spanier, K., Leskovec, J., McLaughlin, C. N., Xie, Q., Jones, R. C., Brueckner, K., Shim, J., Tattikota, S. G., Schnorrer, F., Rust, K., . . . Zinzen, R. P. (2022). Fly Cell Atlas: A single-nucleus transcriptomic atlas of the adult fruit fly. *Science*, 375(6584), eabk2432. <https://doi.org/10.1126/science.abk2432>
- Luu, L. M., Nguyen, L., Peng, S., Lee, J., Lee, H. Y., Wong, C. H., Hergenrother, P. J., Chan, H. Y., & Zimmerman, S. C. (2016). A Potent Inhibitor of Protein Sequestration by Expanded Triplet (CUG) Repeats that Shows Phenotypic Improvements in a Drosophila Model of Myotonic Dystrophy. *ChemMedChem*, 11(13), 1428-1435. <https://doi.org/10.1002/cmdc.201600081>
- Picchio, L., Plantie, E., Renaud, Y., Poovthumkadavil, P., & Jagla, K. (2013). Novel Drosophila model of myotonic dystrophy type 1: phenotypic characterization and genome-wide view of altered gene expression. *Hum Mol Genet*, 22(14), 2795-2810. <https://doi.org/10.1093/hmg/ddt127>
- Sanyal, S. (2009). Genomic mapping and expression patterns of C380, OK6 and D42 enhancer trap lines in the larval nervous system of Drosophila. *Gene Expr Patterns*, 9(5), 371-380. <https://doi.org/10.1016/j.gep.2009.01.002>

- Smith, R. B., Machamer, J. B., Kim, N. C., Hays, T. S., & Marques, G. (2012). Relay of retrograde synaptogenic signals through axonal transport of BMP receptors. *J Cell Sci*, 125(Pt 16), 3752-3764. <https://doi.org/10.1242/jcs.094292>
- Spring, A. M., Brusich, D. J., & Frank, C. A. (2016). C-terminal Src Kinase Gates Homeostatic Synaptic Plasticity and Regulates Fasciclin II Expression at the Drosophila Neuromuscular Junction. *PLoS Genet*, 12(2), e1005886. <https://doi.org/10.1371/journal.pgen.1005886>
- Wright, J. W., & Copenhaver, P. F. (2001). Cell type-specific expression of fasciclin II isoforms reveals neuronal-glia interactions during peripheral nerve growth. *Dev Biol*, 234(1), 24-41. <https://doi.org/10.1006/dbio.2001.0247>
- Zhao, Z., & Geisbrecht, E. R. (2025). Stage-specific modulation of Drosophila gene expression with muscle GAL4 promoters. *Fly (Austin)*, 19(1), 2447617. <https://doi.org/10.1080/19336934.2024.2447617>

REVIEWERS' COMMENTS

Reviewer #1 (Remarks to the Author):

I appreciate the considerable effort the authors have put into revising their manuscript. It is clear that they took the comments seriously and have added a number of new experiments, including electrophysiological analyses, immunostainings, and initial pathway tests. The presentation has also improved, and the revised version reads more clearly than before. I fully acknowledge and value this commitment and the amount of work that went into the revision.

RESPONSE: We thank the reviewer for giving us helpful suggestions in the previous round of revision, which helped us greatly improve the study, and we are very happy that the reviewer acknowledges our effort.

That said, I must admit that I still remain with my main reservations. My initial concerns were not about a single missing experiment, but rather about the overall mechanistic depth of the study. While the new data extend the descriptive part of the work, they do not substantially strengthen the causal or mechanistic link between CUG-repeat toxicity and FasII dysregulation. The proposed pre- and postsynaptic “synergistic” effects also remain interpretative rather than experimentally dissected. In this sense, the revision does not really change the conceptual level of the study.

From my own perspective as someone working on synaptic mechanisms, the study remains primarily correlative and lacks the mechanistic resolution that would turn it into a conceptual advance. It is also a little difficult for me to judge the specific relevance for the DM1 field. If my expert colleagues in that area felt that the work represents a clear and important step forward, I would not object to publication. However, from the viewpoint of synaptic biology, I am sorry to say that I do not see the added value or conceptual advance that would justify publication in Nature Communications.

RESPONSE: We sincerely thank the reviewer for the thoughtful and detailed comments, as well as for highlighting the mechanistic aspects that merit further investigation. We acknowledge that our current study primarily advances descriptive and correlative findings regarding CUG-repeat toxicity and FasII dysregulation, and that additional experiments would be required to fully elucidate the underlying presynaptic and postsynaptic mechanisms. Our intention has been to lay the groundwork for future mechanistic dissection by identifying reproducible synaptic phenotypes and implicating FasII as a potential mediator in the context of DM1 models. We agree that while our data do not yet resolve all causal links, these findings provide a necessary framework and important entry points for further mechanistic studies. Moreover, we believe that the synaptic phenotypes described here, and their possible connection to DM1 pathogenesis, will be of relevance to researchers in both the DM1 field and the wider synaptic biology community, helping to bridge these domains.

We are grateful for the reviewer’s openness to publication if the DM1 community considers these advances significant, and we hope that by situating our work as a foundation for future research, it can be of value to the field. Thank you again for the constructive feedback and consideration.

Reviewer #2 (Remarks to the Author):

Authors in response to my and other reviewer's comments improved significantly their manuscript.

RESPONSE: We thank the reviewer for giving us helpful suggestions that significantly improved our manuscript.

One of important supplementary data provided in the revised version is the analysis of a reduced Mbl level. The authors show that the heterozygous Mbl mutant context mimics toxic CTG repeats and FasII OE effects on NMJ. This finding reinforces authors conclusions. However, when using heterozygous Mbl mutant authors do not address specific pre- versus post-synaptic roles of Mbl attenuation (analysed in the contexts of CTG repeats and Fas2). It would be thus appropriate and of interest to the field to provide this analysis (using UAS-MblRNAi line(s)).

RESPONSE: We agree with the reviewer that RNAi would have provided us with the tools to address the pre-versus post-synaptic roles of Mbl attenuation. However, as Mbl is not our study’s focus, we hope the reviewer would kindly let us round-up the study this time without adding this additional experiment.

Regarding author's response to my first comment and to be more precise: As the DM1 neuropathology includes brain impairment authors need to clarify which aspects of neuropathology they analyse.

One possibility could be by modifying the last sentence in the introduction.

For example:

"These findings provide important new insights into the mechanisms underlying motor defects associated with DM1 neuropathology".

RESPONSE: We thank and agree with the reviewer. We have modified the last sentence in the introduction according to the reviewer's suggestion.

Reviewer #3 (Remarks to the Author):

The authors have made a genuine effort to address the queries of all three reviews. The manuscript is all the better for it.

I still have a question that remains unanswered. The authors should clarify if they think that NCAM1 expression is upregulated in DMSXL and in DM1.

RESPONSE: We thank the reviewer for raising this point and apologise for any confusion. Yes, our data support that NCAM1 expression is upregulated in DMSXL mice and in DM1, although the direction and magnitude of the change vary across tissues. As shown in Figure 3, slot blot analyses revealed a significant increase in total NCAM1 protein in the spinal cord and tibialis anterior of DMSXL mice, as well as in the frontal cortex of patients with DM1 (with expected inter-individual variability). To clarify this in the manuscript, we have revised slightly the Results section as follows:

Page 12: "Using the slot blot technique, we found that NCAM1 **protein** was indeed upregulated in the spinal cord and tibialis anterior of DMSXL mice (Fig. 3i-l), confirming that our findings in *Drosophila* are relevant to mammals."

Page 12: "We again observed the upregulation of NCAM1 **protein** in these human tissues (Fig. 3m and 3n), confirming that our findings are truly relevant to patients with DM1."

In contrast, NCAM1 levels were reduced in DM1 deltoids. As discussed in the manuscript, the slot blot method does not distinguish between isoforms, and NCAM1 regulation may differ by tissue or cell type. A detailed isoform-specific analysis would be required to fully resolve this. The point is discussed in the final section of the manuscript (changes are **bolded** below):

Page 30: "Nevertheless, dysregulation of NCAM1 was still detected in human deltoids and is likely to have resulted in synaptic dysfunction at the NMJ. Alternatively, mature human muscle cells may have compensatory mechanisms that could explain the high level of NCAM1 in transdifferentiated myotubes but low level in deltoid muscles."

In summary, our results consistently show **NCAM1 dysregulation in DM1, with upregulation in DMSXL neural and skeletal muscle tissues and in DM1 frontal cortex, and reduced levels in deltoids**, supporting a model in which tissue-specific NCAM1 changes contribute to DM1 pathogenesis.

The other thing is a technical issue. I am not sure that the way the assay seems to be described for supplemental figures 17 and 18 is the correct way to assess alternatively spliced isoforms of NCAM. It seems like they used primers situated in exon 16 and 17 for the isoform that includes exon 17 and a primer pair located in exon 16 and 18 for the other. But, if exon 17 is alternatively spliced, two products should have shown up in the exon 16-18 product; one representing exons 16,17,18 and the other being exons 16 and 18 (i.e. exon 17 is spliced out). But that is not what the data shows. The way the results appear, it looks like there is no alternative splicing of exon 17. The Sashimi plot confirms that. If you look at it, it looks like there are transcripts that contain exon 16 and 17 that seem to terminate there (i.e. no 3' exons are joined to this transcript), and there are no transcripts that contain exon 16, 17 and 18. This is not an alternative splicing pattern.

If the implication of the data is that MBNL proteins are causing alternative splicing defects in NCAM in the mouse situation, as the MBNL proteins do for 100s of other transcripts in DM1, and that is how we tie the effects on Fas in flies to the spliceopathy model of RNA toxicity in DM1, then the conclusion is incorrect.

Please correct your discussion and don't imply splicing defects in NCAM for the human or mouse situation. Your data doesn't show it.

It is more likely a transcriptional upregulation.

RESPONSE: We thank the reviewer for this valuable and detailed comment, which helped us clarify the nature of the abnormal RNA event observed in *Ncam1* transcripts. Based on the literature and public transcript annotations,

Ncam1 undergoes alternative usage of mutually exclusive terminal (distal) exons, resulting in distinct 3' terminal isoforms. One major isoform includes exon 16 followed by exon 17 only (labelled E16+E17 in Supplementary Fig. 17), while another includes exon 16 followed by a distal block of exons (exons 18–21) (labelled E16+E18+E19+E21). Using a multiplex semi-quantitative RT-PCR strategy with primer pairs spanning exon 16 to either exon 17 or exon 18, we detected a shift in terminal exon usage in both DMSXL mice and DM1 samples. Specifically, the DM1 condition was associated with increased inclusion of distal exons 18, 19 and 21, with exon 20 being excluded in the samples analysed. We agree with the reviewer that this mechanism should be described as altered terminal exon usage rather than “classical” exon skipping of exon 17, and we have revised the text accordingly to avoid any implication of an exon-17 splicing defect that is not supported by our data. The changes are listed below (changes are **bolded** below).

Discussion, page 31-32:

As we have demonstrated that the expression of total NCAM1 is dysregulated in neural and muscular tissues from DMSXL mice (Fig. 3i–l), an important question concerns whether *Ncam1* **RNA isoforms** are also dysregulated in these animals. To answer this question, we subjected primary astrocytes and frontal cortex of DMSXL mice to RT-PCR analysis of **distal exon selection**. As expected, we indeed observed **dysregulated usage of distal *Ncam1* terminal exons** in these cells and tissues (Supplementary Fig. 17). In addition, we examined whether dysregulated *Ncam1* splicing could be caused by *Mbnl1/Mbnl2* double knockdown or knockout. Our results showed that *Ncam1* **terminal exon splicing** was indeed dysregulated in cultured astrocytes subjected to *Mbnl1/Mbnl2* double knockdown (Supplementary Fig. 18a and 18b) and in the frontal cortex of *Mbnl1/Mbnl2* conditional double-knockout mice (*Mbnl1*^{-/-}; *Mbnl2*^{ec}; Nestin-Cre⁺)⁶¹ (Supplementary Fig. 18c and 18d). **As these defects parallel those seen in DMSXL mice (Supplementary Fig. 17), the shift in *Ncam1* terminal exon usage in DM1 is likely linked to functional depletion of MBNL proteins.**

Methods, page 36-37:

Semi-quantitative RT-PCR for the analysis of alternative exon usage. RNA extraction was performed with the RNeasy Mini kit (QIAGEN; 74104) following the manufacturer’s protocol, including a DNase digestion step (RNase-Free DNase Set; QIAGEN; 79254) after the first wash with RW1 buffer. RNA concentration was assessed using the NanoDrop (Thermo Scientific) and RNA quality was verified by electrophoresis on agarose gel. cDNA synthesis and semi-quantitative reverse-transcriptase PCR analysis of alternative **exon usage** were performed as previously described^{66,67}. Oligonucleotide primers used for RT-PCR analysis of *Ncam1* were the following: *Ncam1* E16 (forward): ACGTCATGCTCAAGTCCCTG. *Ncam1* E17 (reverse): AGTCACCGCAGAGAAAAGCA. *Ncam1* E18 (reverse): ATGAGCAGGCCACACTTGTT.

Supplementary figure 17.

Panels b and d. The amplified isoforms are now labelled as **E16+E17** and **E16+E18+E19+E21**.

Panels c and e. The y-axis of the graph now reads: **Percentage distal exon usage**

The legends now reads:

(c) Quantification of the *Ncam1* RNA isoform containing the distal exons E18, E19 and E21 in (b).

(e) Quantification of the *Ncam1* RNA isoform containing the distal exons E18, E19 and E21 in (d).

Supplementary figure 18.

Panels a and c. The amplified isoforms are now labelled as **E16+E17** and **E16+E18+E19+E21**.

Panels b and d. The y-axis of the graph now reads: **Percentage distal exon usage**

The legends now reads:

(b) Quantification of the *Ncam1* RNA isoform containing the distal exons E18, E19 and E21 in (a).

(d) Quantification of the *Ncam1* RNA isoform containing the distal exons E18, E19 and E21 in (c).

Otherwise, you have done a commendable job and I have no other issues.

RESPONSE: We greatly appreciate the reviewer for giving us helpful suggestions to improve our study.